# Food and water intake are regulated by distinct central amygdala circuits revealed using intersectional genetics

Federica Fermani[1], Simon Chang [2], Ylenia Mastrodicasa [1], Christian Peters [1], Louise Gaitanos[1], Pilar L. Alcala Morales[1], Charu Ramakrishnan [3], Karl Deisseroth [3,4,5] & Rüdiger Klein [1] ✉

The central amygdala (CeA) plays a crucial role in defensive and appetitive behaviours. It contains genetically defined GABAergic neuron subpopulations distributed over three anatomical subregions, capsular (CeC), lateral (CeL), and medial (CeM). The roles that these molecularly- and anatomically-defined CeA neurons play in appetitive behavior remain unclear. Using intersectional genetics in mice, we found that neurons driving food or water consumption are confined to the CeM. Separate CeM subpopulations exist for water only versus water or food consumption. In vivo calcium imaging revealed that CeM[Htr2a] neurons promoting feeding are responsive towards appetitive cues with little regard for their physical attributes. CeM[Sst] neurons involved in drinking are sensitive to the physical properties of salient stimuli. Both CeM subtypes receive inhibitory input from CeL and send projections to the parabrachial nucleus to promote appetitive behavior. These results suggest that distinct CeM microcircuits evaluate liquid and solid appetitive stimuli to drive the appropriate behavioral responses.

The central amygdala (CeA), is a brain hub for emotion and motivation that rapidly integrates salient environmental and internal stimuli to generate appropriate behavioral responses[1–4]. While traditionally thought to be linked solely to defensive behavior, emerging evidence suggests that specific CeA neuronal populations play key roles in reward and consummatory behaviors, such as feeding[5,6] and drinking[7]. Since food and water are intrinsically rewarding[8,9], appetitive CeA neurons are positive valence neurons and the animals seek to promote the activation of these neurons. In vivo recordings have shown that appetitive CeA neurons respond to appetitive stimuli, such as food and water, as well as to cues predictive of these stimuli[1,2,5,10–12]. In vivo single-cell calcium imaging experiments revealed marked heterogeneity in specific appetitive subpopulations, with some appetitive neurons becoming selectively activated by appetitive cues, with others

becoming inhibited or showing dynamic responses to multiple stimuli[5,12,13]. In addition to environmental stimuli, appetitive CeA neurons are activated by internal signals, such as the hunger hormone ghrelin[13]. Despite recent progress, the circuit mechanisms in the central amygdala that process appetitive signals such as food and water are still not well understood[2,14–17].

The CeA consists exclusively of γ-aminobutyric acid-releasing (GABAergic) neurons that can be divided into subpopulations based on their molecular, morphological, physiological[18], and functional properties. They are organized in inhibitory microcircuits in three subregions of the CeA: the central capsular (CeC), lateral (CeL), and medial subregions (CeM). For defensive behavior, the information flow through the subregions has been partially characterized: CeC and CeL subregions are the primary targets for sensory inputs that are

[1]Department of Molecules – Signaling – Development, Max-Planck Institute for Biological Intelligence, Martinsried, Germany. [2]Cellular Neurobiology, Department of Behavioral and Molecular Neurobiology, University of Regensburg, Regensburg, Germany. [3]Department of Bioengineering, Stanford University, Stanford, CA, USA. [4]Department of Psychiatry & Behavioral Sciences, Stanford University, Stanford, CA, USA. [5]Howard Hughes Medical Institute, Stanford University, Stanford, CA, USA. ✉e-mail: ruediger.klein@bi.mpg.de

processed and passed on to the CeM from where major projections go to hindbrain autonomic and motor control areas[2,19,20]. For appetitive behavior, the information flow is less clear. New perspectives have been suggested, with the CeL also forming long-range efferent projections bypassing the CeM, and the CeM receiving direct inputs bypassing the CeL[21,22].

The functional characterization of molecularly-defined appetitive CeA subpopulations has been the subject of recent intense research. It has mostly relied on the manipulation of single Cre driver lines and expression analysis of selected markers[5,7,23–25]. The more recent characterization of CeA subpopulations by scRNAseq analysis revealed that several of these Cre drivers are expressed in more than one cell cluster and, importantly, across CeA subdivisions[13,22,26]. The Sst-Cre driver has been extensively used to demonstrate that CeA neurons expressing the neuropeptide somatostatin, control both appetitive and aversive behaviors[7,11,12,27,28]. In vivo calcium imaging suggested that Sst+ CeA neurons help discriminate between stimuli with different physical properties and participate in the evaluation of salient events during reward learning[12]. CeA$^{Sst}$ neurons can be found in CeL and CeM subregions and partially overlap with neurons expressing neurotensin (Nts)[7,29] which show a similar distribution pattern and promote consumption of palatable fluids[24]. The anatomical position of appetitive CeA$^{Sst}$ and CeA$^{Nts}$ neurons has not been fully unraveled.

In previous work, we have shown that CeA neurons expressing the Htr2a-Cre driver promote food consumption through a positive valence signal[5]. CeA$^{Htr2a}$ neurons are activated by fasting, the hunger hormone ghrelin, and the presence of food[13]. Single-cell transcriptomics revealed that CeA$^{Htr2a}$ neurons are located in the CeL and CeM and it is currently unclear which of these populations promotes food consumption. A large fraction of Htr2a-Cre-expressing CeA neurons overlaps with neurons expressing prepronociceptin (Pnoc)[13]. Similar to CeA$^{Htr2a}$ neurons, CeA$^{Pnoc}$ neurons promote palatable food consumption[23], but the exact anatomical location of Pnoc-positive appetitive neurons remains unclear. CeA$^{Htr2a}$ neurons located in the CeL overlap with the Sst-positive population, whereas CeM$^{Htr2a}$ and CeM$^{Sst}$ neurons are largely separate populations[13].

To better define the CeA appetitive microcircuits, we employed combinatorial recombinase-dependent targeting[30] using available Cre lines in combination with a transgenic line that expresses the Flp recombinase specifically in neurons of the CeL subregion. This approach, in combination with INTRSECT Boolean vectors[31,32], allowed for functional characterization of four spatially and molecularly distinct CeA subpopulations, CeL$^{Sst}$, CeM$^{Sst}$, CeL$^{Htr2a}$, and CeM$^{Htr2a}$ neurons. The results revealed the organization of appetitive information flow through the CeA. Our results indicate that neurons driving food or water consumption are confined to the CeM. Separate CeM subpopulations exist for water only (CeM$^{Sst}$), and water or food consumption (CeM$^{Htr2a}$). In vivo calcium imaging revealed that CeM$^{Htr2a}$ neurons promoting feeding or drinking are highly responsive towards appetitive cues with little regard for their physical attributes. CeM$^{Sst}$ neurons involved exclusively in drinking are sensitive to the physical properties of salient stimuli. Both CeM subtypes are controlled by inhibitory signals from the CeL and, in turn, form long-range inhibitory projections to the PBN to promote water or food consumption. These results suggest that distinct CeM microcircuits evaluate liquid and solid appetitive stimuli to drive the appropriate behavioral responses.

## Results

### A FlpoER transgenic line for intersectionally targeting central amygdala neurons

To manipulate CeL versus CeM neurons with an intersectional genetics strategy, we generated a Wfs1-FlpoER mouse line that expresses the tamoxifen-inducible optimized FlpoER recombinase under the control of the Wolframin1 (Wfs1) promoter (Fig. 1a). Wfs1 was previously shown to mark the majority of cells in the CeL[33,34] and we confirmed

that Wfs1 immunoreactivity was enriched in the CeL (Fig. 1b). Using Htr2a-Cre and Sst-Cre driven tdTomato reporter mice, we found that nearly 90% of Htr2a-Cre and Sst-Cre-positive cells expressed Wfs1 in the CeL, and a large fraction of Wfs1 immunoreactive cells encompassed Htr2a-Cre and Sst-Cre-positive cells (Supplementary Fig. 1a–j). To validate the accuracy of FlpoER expression, we bred Wfs1-FlpoER mice with an FPDI reporter line which expresses mCherry after Flp-mediated recombination[35]. After three tamoxifen injections, 56% of cells in CeL that were immunopositive for endogenous Wfs1 expressed FlpoER (mCherry), and, importantly for our intersectional targeting strategy, mCherry+ cells were 17-times more abundant in CeL than CeM (Supplementary Fig. 1k–o).

Crossing Htr2a-Cre or Sst-Cre with Wfs1-FlpoER mice generated double transgenic "intersectional mice" (Htr2a-Cre;Wfs1-FlpoER and Sst-Cre;Wfs1-FlpoER), in which most cells in the CeL should express both Cre and FlpoER, while the vast majority of cells in the CeM should only express Cre and not FlpoER (Fig. 1c). Injection of INTRSECT Boolean vectors[31,32] into the CeA of these mice made it possible to specifically manipulate either the CeL (CreON/FlpON virus or Con/Fon in short) or the CeM population (CreON/FlpOFF virus or Con/Foff) (Fig. 1d). To validate the fidelity of this approach, we injected either Con/Fon or Con/Foff EYFP virus into the CeA of Htr2a-Cre;Wfs1-FlpoER mice (together with a Cre-dependent mCherry virus to visualize the entire Cre-positive population). The results with the Con/Fon-EYFP virus showed that EYFP+ cells were more abundant in the CeL than CeM across the anterior-posterior extent of the CeA (Fig. 1e, f). Quantifications revealed that among the Htr2a-Cre-positive cells in the CeL, the majority (54%) expressed EYFP (Fig. 1g) and 4-times as many EYFP cells (80%) were present in CeL versus CeM (Fig. 1h). Conversely, with the Con/Foff virus, EYFP+ cells were more abundant in the CeM than the CeL (Fig. 1i, j). Quantifications revealed that among the Htr2a-Cre-positive cells in the CeM, the majority (66%) expressed EYFP (Fig. 1k), and 2.3-times as many EYFP cells (70%) were present in CeM versus CeL (Fig. 1l). Similar results were obtained with Sst-Cre;Wfs1-FlpoER mice. With the Con/Fon-EYFP virus, EYFP+ cells were more abundant in the CeL than CeM (Fig. 1m, n). Among Sst-Cre-positive cells in the CeL a large fraction (48%) expressed EYFP (Fig. 1o) and 10 times as many EYFP cells (91%) were present in CeL versus CeM (Fig. 1p). With the Con/Foff virus, EYFP+ cells were more abundant in the CeM than CeL (Fig. 1q, r). Among the Sst-Cre-positive cells in the CeM the majority (69%) expressed EYFP (Fig. 1s) and 2.7 times as many EYFP cells (71%) were present in CeM versus CeL (Fig. 1t). These experiments show that our Cre/Flp intersectional mouse model, combined with INTRSECT viruses, enabled an enrichment of actuator expression in a specific subpopulation within a single CeA subregion.

### Promotion of water intake exclusively by CeM subpopulations

To study the role of different CeA subpopulations in water consumption, we first asked if optogenetic activation of the neurons would be sufficient to promote water intake. We stereotactically injected the Con/Fon- or Con/Foff-hChR2-EYFP viruses bilaterally into the CeA of Htr2a-Cre;Wfs1-FlpoER mice to express channelrhodopsin (ChR2) in the CeL and CeM subpopulations, respectively (Fig. 2a). Similar INTRSECT viruses expressing only EYFP control protein were used as negative controls. As positive controls, we expressed ChR2 in the entire CeA$^{Htr2a}$ population by including one cohort of Htr2a-Cre mice that was injected with the simple Cre-dependent DIO-hChR2-EYFP virus (Fig. 2a). Similar injections were performed with Sst-Cre;Wfs1-FlpoER and Sst-Cre mice to analyze the CeA$^{Sst}$ subpopulations. The expression of EYFP in different CeA subregions was validated in all animals (Fig. 2b–e). Optic fibers were placed bilaterally over the CeA to photoactivate the cell bodies of the neurons (Supplementary Fig. 2a–g). In a 30-min paradigm, thirsty animals were exposed to water while being bilaterally photoactivated throughout the session. The same experiment was repeated when animals were hydrated. We

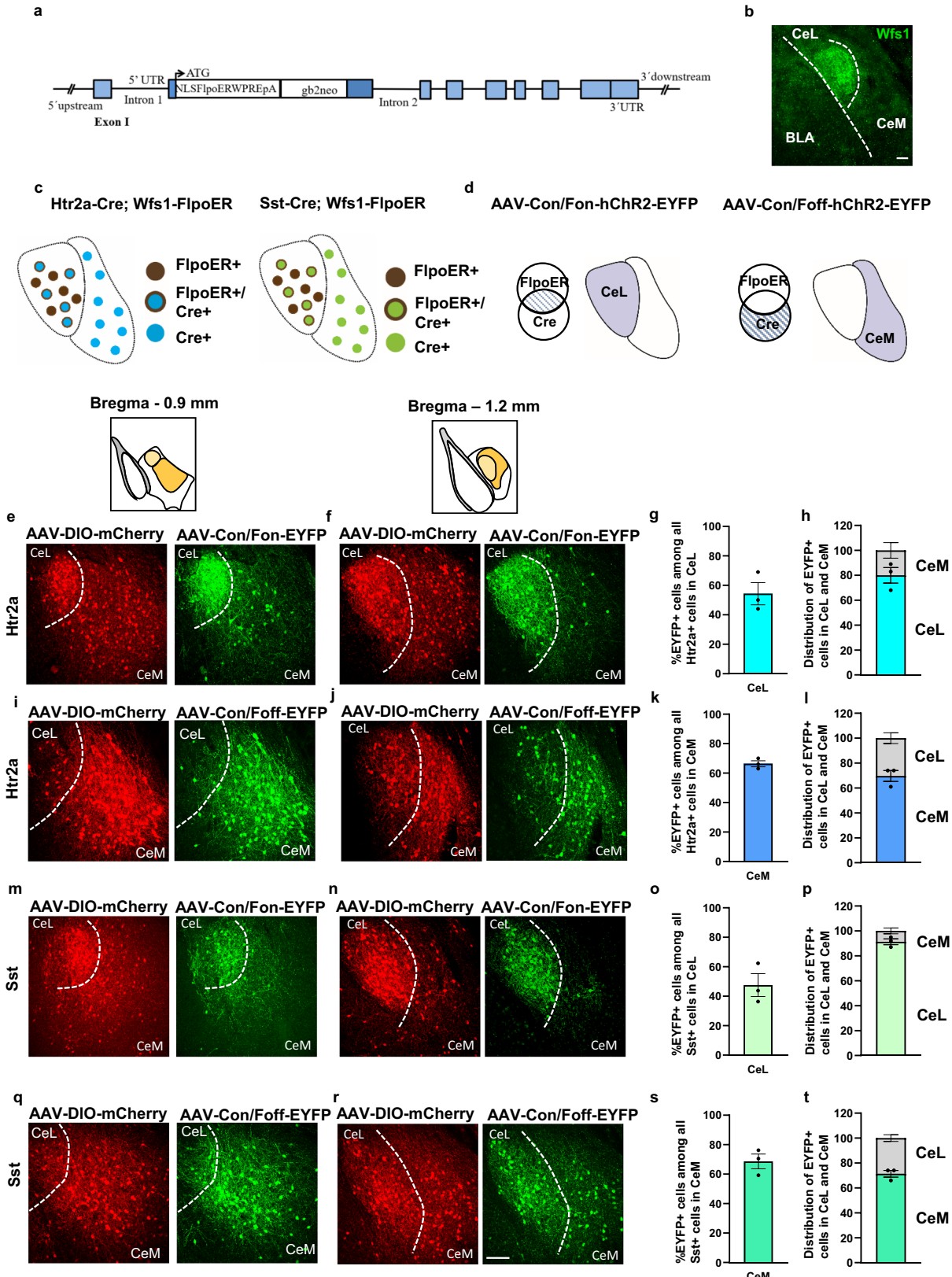

found that photoactivation of the entire CeA$^{Htr2a}$ and CeA$^{Sst}$ populations was sufficient to promote water consumption by thirsty as well as hydrated mice (Fig. 2f; Supplementary Fig. 3a). Surprisingly, among the two subpopulations, it was exclusively the CeM fractions of both cell types that promoted water intake both in thirsty and hydrated mice (Fig. 2g, h; Supplementary Fig. 3b, c).

To ensure consistency in the effects of optogenetic stimulation across multiple trials, we used a different drinking test during which short (10-min) periods of photoactivation alternated with periods of no photoactivation. During an initial 10-min Light-OFF period, water-deprived mice from all experimental groups were found to consume similar amounts of water (Fig. 2i–l). As the experiment progressed,

**Fig. 1 | Characterization of the CeL-specific Wfs1-FlpoER driver line. a** Wfs1-FlpoER BAC transgene. Abbreviations: FlpoER optimized Flp-estrogen receptor fusion protein, NLS nuclear localization sequence, WPRE-pA woodchuck hepatitis post-transcriptional regulatory element with bovine growth hormone poly-adenylation signal, gb2neo gb2 prokaryotic promoter driven Neo cassette. **b** Example showing endogenous Wfs1 immunoreactivity in the CeL. **c** Intersectional strategy used with Htr2a-Cre;Wfs1-FlpoER and Sst-Cre;Wfs1-FlpoER mice. Cells in CeL are either FlpoER positive or FlpoER and Cre double positive. Cells in CeM are only Cre-positive. **d** Examples of INTRSECT viruses. Con/Fon viruses express in CeL cells that are double positive for Cre and FlpoER, while Con/Foff viruses express in CeM cells that are Cre-positive and FlpoER-negative. **e, t** Expression through anterior-posterior axis [Bregma −0.9 (**e, i, m, q**) and −1.2 (**f, j, n, r**)] of Cre-dependent AAV-DIO-mCherry (not intersectional, red) in Htr2a-Cre;Wfs1-FlpoER mice showing the entire population of CeA$_{Htr2a}$ cells in the CeL and CeM (**e, f**), and expression of AAV-Con/Fon-EYFP (green) in the same mice predominantly highlighting the CeL fraction of Htr2a positive cells (**e, f**).

Quantifications of the fraction of Htr2a-Cre::Con/Fon-EYFP positive cells in the CeL (54.3 ± 7.5%) among all CeA$^{Htr2a}$ cells in CeL (**g**) and distribution of Htr2a-Cre;Con/Fon-EYFP positive cells in CeL and CeM (**h**). **i, j** Expression of AAV-Con/Foff-EYFP (CeM enriched, green) versus AAV-DIO-mCherry (red) as described for (**e, f**). Quantifications of the fractions of Htr2a-Cre::Con/Foff-EYFP positive cells in CeM (66.3 ± 2.0%) among all CeA$^{Htr2a}$ cells in CeM (**k**) and distribution of Htr2a-Cre;Con/Foff-EYFP positive cells in CeL and CeM (**l**). **m, n** Expression of AAV-Con/Fon-EYFP (CeL enriched, green) versus AAV-DIO-mCherry (red) in Sst-Cre;Wfs1-FlpoER mice. **o, p** Quantification of the fraction of Sst-Cre::Con/Fon-EYFP positive cells in CeL (47.5 ± 7.7%) among all Sst-positive cells in CeL (**o**) and distribution of Sst-Cre;Con/Fon-EYFP positive cells in CeL and CeM (**p**). **q, r** Expression of AAV-Con/Foff-EYFP (CeM enriched, green) versus AAV-DIO-mCherry (red) in Sst-Cre;Wfs1-FlpoER mice. Quantification of the fraction of Sst-Cre::Con/Foff-EYFP positive cells in CeM (68.6 ± 5%) among all CeA$^{Sst}$ cells in CeM (**s**) and distribution of Sst-Cre;Con/Foff-EYFP positive cells in CeL and CeM (**t**) (all groups: n = 3 mice/9 sections). Values = Mean ± SEM. Scale bar: 115 μm.

control animals gradually became hydrated and consumed less water, independent of whether the light was ON or OFF. Among the experimental groups, only photoactivation of the entire CeA populations or the CeM subpopulations increased water consumption during each Light-ON period (Fig. 2i, j; Supplementary Fig. 3d, e). Photoactivation of the CeL subpopulations was insufficient to promote water consumption (Fig. 2k, l). Similar results were obtained with hydrated animals (Supplementary Fig. 3f–k). Overall consumption of water was highest during Light-ON periods for mice expressing ChR2 in the entire CeA or CeM subpopulations of Htr2a$^+$ and Sst$^+$ cells, while water consumption was unaffected in photoactivated controls and mice expressing ChR2 in the CeL (Supplementary Fig. 4). In summary, these experiments showed that the CeM subpopulations of Htr2a-Cre and Sst-Cre-positive cells were sufficient to drive water consumption.

We next asked if the CeM subpopulations were also required for water consumption. We stereotactically injected AAVs expressing Cre-dependent eNpHR3.0 bilaterally into the CeA of Htr2a-Cre mice to express the inhibitory Halorhodopsin in the entire CeA$^{Htr2a}$ population (Fig. 3a), or similarly INTRSECT Con/Foff IC++, a blue-shifted chloride-conducting channelrhodopsin[32,36], into Htr2a-Cre;Wfs1-FlpoER mice to express the actuator specifically in the CeM subpopulation (Fig. 3b). Similar injections were performed with Sst-Cre;Wfs1-FlpoER mice to analyze the CeM$^{Sst}$ subpopulation (Fig. 3a, b). Corresponding Cre-dependent and INTRSECT viruses expressing mCherry or EYFP control proteins were injected as controls (Fig. 3a, b). Animals were water deprived and tested for water consumption for 30 min while being photoinhibited. Inhibition of the entire CeA population, as well as the respective CeM$^{Htr2a}$ and CeM$^{Sst}$ subpopulations significantly decreased water intake compared to light-stimulated controls (Fig. 3c, d). Together, these experiments reveal that the CeM subpopulations of Htr2a-Cre and Sst-Cre-positive neurons are necessary for efficient water consumption in thirsty mice.

### Promotion of food intake exclusively by CeM$^{Htr2a}$, but not CeM$^{Sst}$, neurons

Next, we focused our attention on feeding behavior. We previously reported that CeA$^{Htr2a}$ neurons stimulated food intake[5,13], but the specific contributions of the CeL and CeM subpopulations had not been explored. Likewise, the roles of the different Sst subtypes in regulating feeding are not completely understood[7]. We performed a free-feeding assay with *ad libitum* fed animals expressing ChR2 in Htr2a-Cre$^+$ or Sst-Cre$^+$ neurons in the CeA, CeL, or CeM. As expected, photoactivation of the entire CeA$^{Htr2a}$ neuron population caused a significant increase in food intake compared to the photoactivated control group (Fig. 4a). Surprisingly, food intake was exclusively driven by the CeM$^{Htr2a}$, but not the CeL$^{Htr2a}$, subpopulation (Fig. 4b, c). Photoactivation of the entire CeA$^{Sst}$ neuron population or its subpopulations in the CeL and CeM

failed to significantly increase food consumption under these conditions (Fig. 4a–c).

To investigate if the activity of the CeM$^{Htr2a}$ subpopulation was required for efficient food consumption, we asked if photoinhibition of CeM$^{Htr2a}$ neurons would reduce consumption of a palatable liquid reward (Fresubin, 2 kcal/ml) as previously shown for the entire CeA$^{Htr2a}$ population[5]. Indeed, photoinhibition of the entire CeA$^{Htr2a}$ population or the CeM$^{Htr2a}$ subpopulation significantly reduced the consumption of the palatable liquid compared to the control groups (Fig. 4d, e). The effect size of photoinhibiting the CeM$^{Htr2a}$ subpopulation was larger compared to the entire CeA$^{Htr2a}$ population, with a Cohen's d of 0.8, indicating a medium effect. In contrast, inhibition of the entire CeA$^{Sst}$ population or the CeM$^{Sst}$ subpopulation did not have a significant effect on Fresubin consumption (Fig. 4d, e). These results provide genetic evidence for CeM$^{Htr2a}$ neurons promoting water and food consumption, whereas CeM$^{Sst}$ neurons had a more restricted role in water, but not food consumption. The CeL subpopulations of Htr2a-Cre$^+$ and Sst-Cre$^+$ neurons, which highly overlap, do not seem to affect water or food consumption, at least in the assays tested.

### CeM subpopulations drive real-time place preference and conditioned reward behavior

Next, we asked which subpopulations could drive innate rewarding, but non-consummatory behavior, in a real-time place preference (RTPP) assay consisting of a two-chamber arena with one compartment paired with laser photostimulation (Fig. 5a). The results revealed a similar pattern as for water intake: activation of the entire populations of CeA$^{Htr2a}$ or CeA$^{Sst}$ neurons as well as the respective CeM subpopulations resulted in the animals exhibiting a significant preference for the photoactivation-paired chamber, as compared to the CeL subpopulations and the controls (Fig. 5b–d). The time spent in the center of an open-field arena as well as the distance moved was unchanged, suggesting that the reward behavior in the RTPP assay was not due to an anxiolytic effect (Supplementary Fig. 5a–e).

Further, we investigated which subpopulation might condition a preference for a specific flavor. In a reward conditioning paradigm, pairing activation of appetitive CeA neurons with one of two flavors can reverse flavor preference, such that an initially less preferred flavor becomes the preferred one[5,17]. Mice expressing ChR2 in the entire populations of CeA$^{Htr2a}$ or CeA$^{Sst}$ neurons, or the respective CeM subpopulations, were allowed to consume two differently flavored non-nutritive liquids. After determining their individual baseline preference, conditioning was performed by pairing the less preferred flavor with optogenetic activation (Fig. 5e). After conditioning, the flavor preference of the mice was assessed by simultaneously offering liquids of both flavors. The results showed that activation of the CeM subpopulations of CeA$^{Htr2a}$ or CeA$^{Sst}$ neurons, reversed the animals' initial preference, resulting in the least preferred flavor becoming the

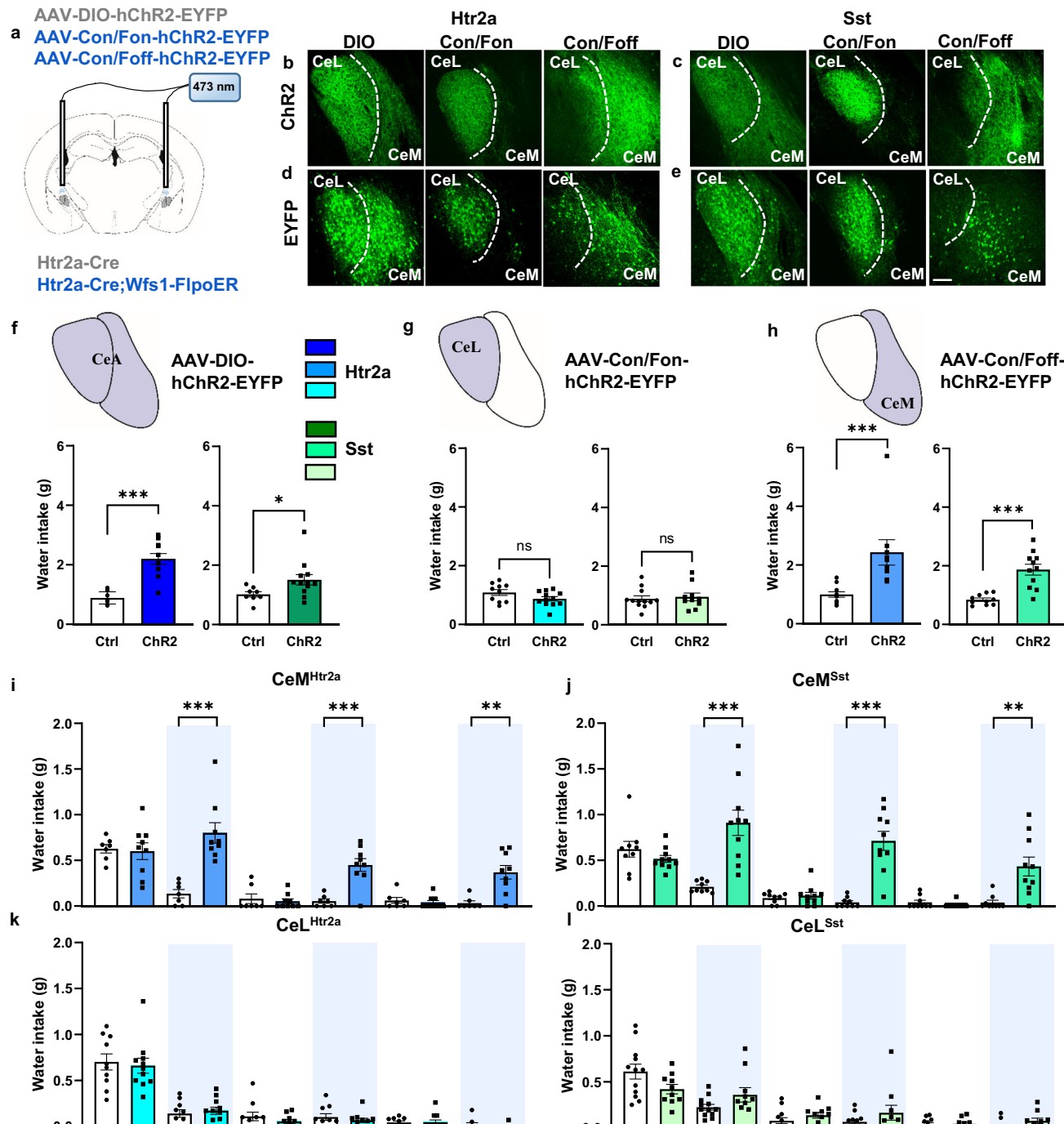

**Fig. 2 | CeM^Htr2a and CeM^Sst neurons promote water intake. a** Viruses injected and optic fiber placement in the CeA (modified from Allen Mouse Brain Atlas, mouse.brain-map.org). Representative images of ChR2-EYFP expression in CeA subregions after injections of INTRSECT Con/Fon or Con/Foff viruses into Htr2a-Cre;Wfs1-FlpoER (**b**) or Sst-Cre;Wfs1-FlpoER mice (**c**). All mice used for behavior showed similar results. **d**, **e** Similar manipulations as in (**b**, **c**) except for INTRSECT expression of EYFP protein. **f** Water intake during photoactivation of all CeA^Htr2a or CeA^Sst neurons for 30-min by water-deprived mice compared to photoactivated controls (Htr2a: unpaired two-tailed t test $p < 0.0001$, t = 6.429. Sst: Mann-Whitney two-tailed U test $p = 0.0124$, U = 16). (CeA^Htr2a: n = 9 Ctrl, n = 11 ChR2. CeA^Sst: n = 8 Ctrl, n = 12 ChR2 mice). **g** Water intake during photoactivation of the CeL (Htr2a: unpaired two-tailed t test $p = 0.0913$, t = 1.774. Sst: unpaired two-tailed t test $p = 0.7017$ t = 0.3886). (CeL^Htr2a: n = 10 Ctrl, n = 12 ChR2. CeL^Sst: n = 12 Ctrl, n = 10 ChR2 mice). **h** Water intake during photoactivation of the CeM (Htr2a: Mann-Whitney two-tailed U test $p < 0.0001$, U = 2. Sst: unpaired two-tailed t test $p < 0.0001$ t = 5.054). (CeM^Htr2a: n = 11 Ctrl, n = 9 ChR2. CeM^Sst: n = 10 Ctrl, n = 11

ChR2 mice). **i–l** Photoactivation of CeM^Htr2a (**i**) or CeM^Sst neurons (**j**) during alternating 10-min light OFF and ON epochs in water-deprived mice increased water intake during light ON epochs (Htr2a: 10–20 min: Mann-Whitney two-tailed U test $p = 0.0002$, U = 0. 30–40 min: unpaired two-tailed t test $p = 0.0003$, t = 4.801. 50–60 min: Mann-Whitney two-tailed U test $p = 0.0030$, U = 5.500. Sst: 10–20 min: unpaired two-tailed t test $p = 0.0002$, t = 4.738. 30–40 min: Mann-Whitney two-tailed U test $p < 0.0001$, U = 1.500. 50–60 min: Mann-Whitney two-tailed U test $p = 0.0012$, U = 8). Photoactivation of the CeL fractions (**k**, **l**) had no effect (Htr2a: 10–20 min: unpaired two-tailed t test $p = 0.6180$, t = 0.5069. 30–40 min: Mann-Whitney two-tailed U test $p = 0.6326$, U = 48. 50–60 min: Mann-Whitney two-tailed U test $p = 0.4624$, U = 49. Sst: 10–20 min: unpaired two-tailed t test $p = 0.1053$, t = 1.697. 30–40 min: Mann-Whitney two-tailed U test $p = 0.4264$, U = 48. 50–60 min: Mann-Whitney two-tailed U test $p = 0.0697$, U = 36). (CeM^Htr2a: n = 7 Ctrl, n = 9 ChR2. CeM^Sst: n = 9 Ctrl, n = 10 ChR2. CeL^Htr2a: n = 10 Ctrl, n = 11 ChR2. CeL^Sst: n = 12 Ctrl, n = 10 ChR2 mice). Values = Mean ± SEM. Scale bar: 115 μm.

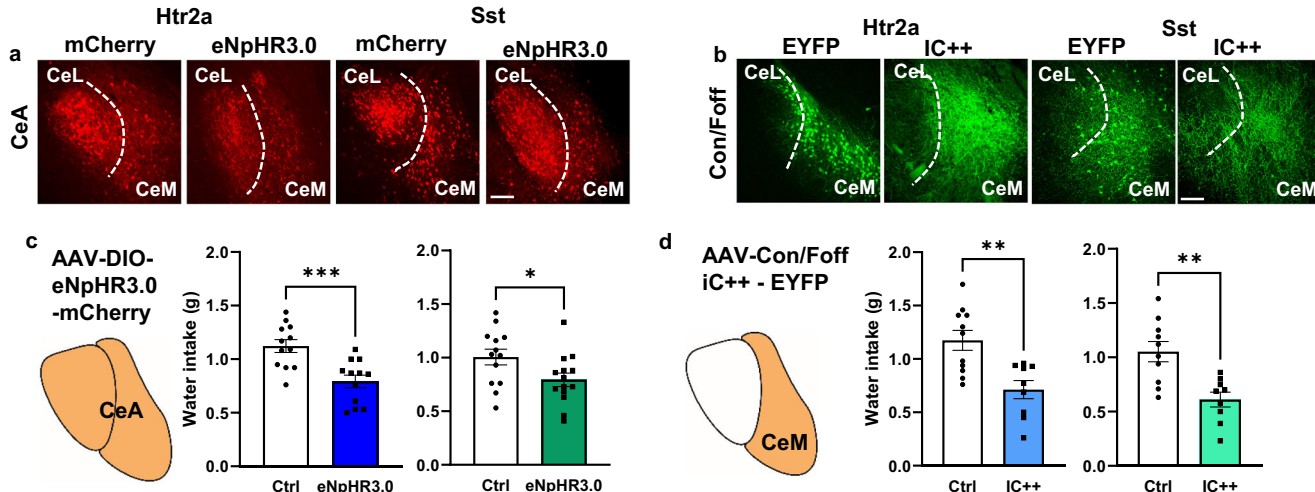

**Fig. 3 | CeM^Htr2a and CeM^Sst neurons are required for normal water intake.**
**a** Representative images showing expression of Cre-dependent eNpHR3.0 and
control mCherry in the entire CeA of Htr2a-Cre and Sst-Cre animals. All mice used
for behavior showed similar results. **b** Representative images showing expression
of IC++ and control YFP predominantly in the CeM after injection of Con/Foff virus
in Htr2a-Cre;Wfs1-FlpoER and Sst-Cre;Wfs1-FlpoER mice. All mice used for behavior
showed similar results. **c** Photoinhibition of the entire populations of CeA^Htr2a (blue)
or CeA^Sst neurons (green) in water-deprived mice significantly decreased water
consumption compared to photoinhibited controls during a 30-min drinking assay

(Htr2a: unpaired two-tailed t test $p = 0.0007$, t = 3.939. Sst: unpaired two-tailed t
test $p = 0.0363$, t = 2.213). (CeA^Htr2a: n = 12 Ctrl, n = 12 eNpHR3.0. CeA^Sst: n = 13 Ctrl,
n = 14 eNpHR3.0 mice). **d** Photoinhibition of the CeM fractions of CeA^Htr2a (blue) or
CeA^Sst neurons (green) significantly decreased water consumption compared to
photoinhibited controls in the same assay (Htr2a: unpaired two-tailed t test
$p = 0.0022$, t = 3.578. Sst: unpaired two-tailed t test $p = 0.0017$, t = 3.733). (CeM^Htr2a:
n = 12 Ctrl, n = 12 IC++. CeM^Sst: n = 11 Ctrl, n = 9 IC++ mice). Values = Mean ± SEM.
Scale bar: 115 μm.

preferred one (Fig. 5f–h). Somewhat surprisingly, also the CeL sub-
populations showed significant conditioning activities (Fig. 5g). Con-
sistent with these results, mice expressing ChR2 in the CeM
subpopulations of CeA^Htr2a or CeA^Sst neurons consumed a significantly
larger amount of the liquid that was paired with photoactivation
compared to controls (Supplementary Fig. 5f–h). These results
demonstrate that the activities of both CeM^Htr2a and CeM^Sst sub-
populations are intrinsically positively reinforcing in RTPP and condi-
tioned flavor preference assays. Both, CeL^Htr2a and CeL^Sst
subpopulations displayed modest reinforcing activity in the condi-
tioned flavor preference assay, but perhaps not in real-time place
preference.

### CeM^Htr2a and CeM^Sst neuron responses to different rewarding behaviors

To understand how CeM^Htr2a and CeM^Sst neurons participate in con-
summatory behavior, we performed single-cell resolution in vivo cal-
cium imaging in freely behaving mice. We injected a Con/Foff-
GCaMP6m virus unilaterally into the CeA of either Htr2a-Cre::Wfs1-
FlpoER or Sst-Cre::Wfs1-FlpoER animals to record the neuronal activity
preferentially from CeM subpopulations. A gradient-index (GRIN) lens
was implanted to monitor the neuronal activity using a head-mounted
miniscope. We tested the animals in three separate conditions: expo-
sure to water, food, and Fresubin. To make consumption of water and
food pleasant, mice were water- and food-deprived, respectively. Fre-
subin instead is very palatable independent of the hunger state. Hence,
mice exposed to Fresubin were fed *ad libitum*. Recordings were done
for 10 min following a 10 min habituation period (Fig. 6a, b). During
habituation, we recorded the activities of an average of 128 neurons
per subpopulation, while during stimulus exposure, the numbers of
active neurons increased to an average of 160 neurons (Supplementary
Fig. 6a). The average number of neurons that were active across dif-
ferent rewarding conditions ranged from 78 to 114 (Supplementary
Fig. 6b). When comparing the activities of all active cells during habi-
tuation and stimulation (including active and inactive times),
we observed a general increase in neuronal activity for both CeM^Htr2a

and CeM^Sst neurons during stimulus exposure, except for CeM^Htr2a
neurons exposed to water (Fig. 6c, d). When comparing the activities
during reward consumption versus inactive episodes during the 10-
min stimulation periods, we found that CeM^Htr2a and CeM^Sst neurons
showed higher activity during drinking and feeding (Supplemen-
tary Fig. 6c).

To investigate whether consummatory behaviors contributed to
the dynamics of neural activity, we analyzed the data using the recently
developed machine learning algorithm CEBRA[37] (for further details,
see "Methods"). We applied the so-called Hybrid model, which con-
siders the association between consumption bouts and the patterns of
neural activity across time. If a behavior contributed significantly to
neural activity, we expected to see clear structure in CEBRA-generated
3D neural embeddings compared to shuffled embeddings. When
applying CEBRA to both CeM^Htr2a and CeM^Sst calcium recordings during
drinking, feeding, and Fresubin consumption, we found clear struc-
tures for all three behaviors, i.e., a separation of data points derived
from times when the animals were engaged in consummatory behavior
from times when the animals were not involved in consumption
(Fig. 6e). Notably, this separation was lost when the data was shuffled,
suggesting contributions of these behaviors to neural activity (Fig. 6e).
To analyze how well the neural embeddings decoded specific beha-
vioral features, we applied a Random Forest (RF) classifier for beha-
vioral decoding (see "Methods"). The results indicated that the
CeM^Htr2a and CeM^Sst neural embeddings decoded the respective
behaviors with 85–90% accuracy, significantly better than shuffled
data (60–75%) (Fig. 6f, g). To measure the prediction error of the RF
classifier, we calculated the out of bag (OOB) error, and found that in
all cases the OOB error from the actual data was much lower than from
shuffled data (Supplementary Fig. 6d). Next, we asked if neurons
associated with different behaviors were preserved across contexts
with shared activity, using longitudinal registration of recorded neu-
rons. We applied CEBRA to neural activity data of neurons shared
between two behaviors (e.g., drinking/feeding, drinking/Fresubin). We
found that the neurons associated with both behaviors showed clearly
structured CEBRA embeddings (Fig. 6h, Supplementary Fig. 6e). To

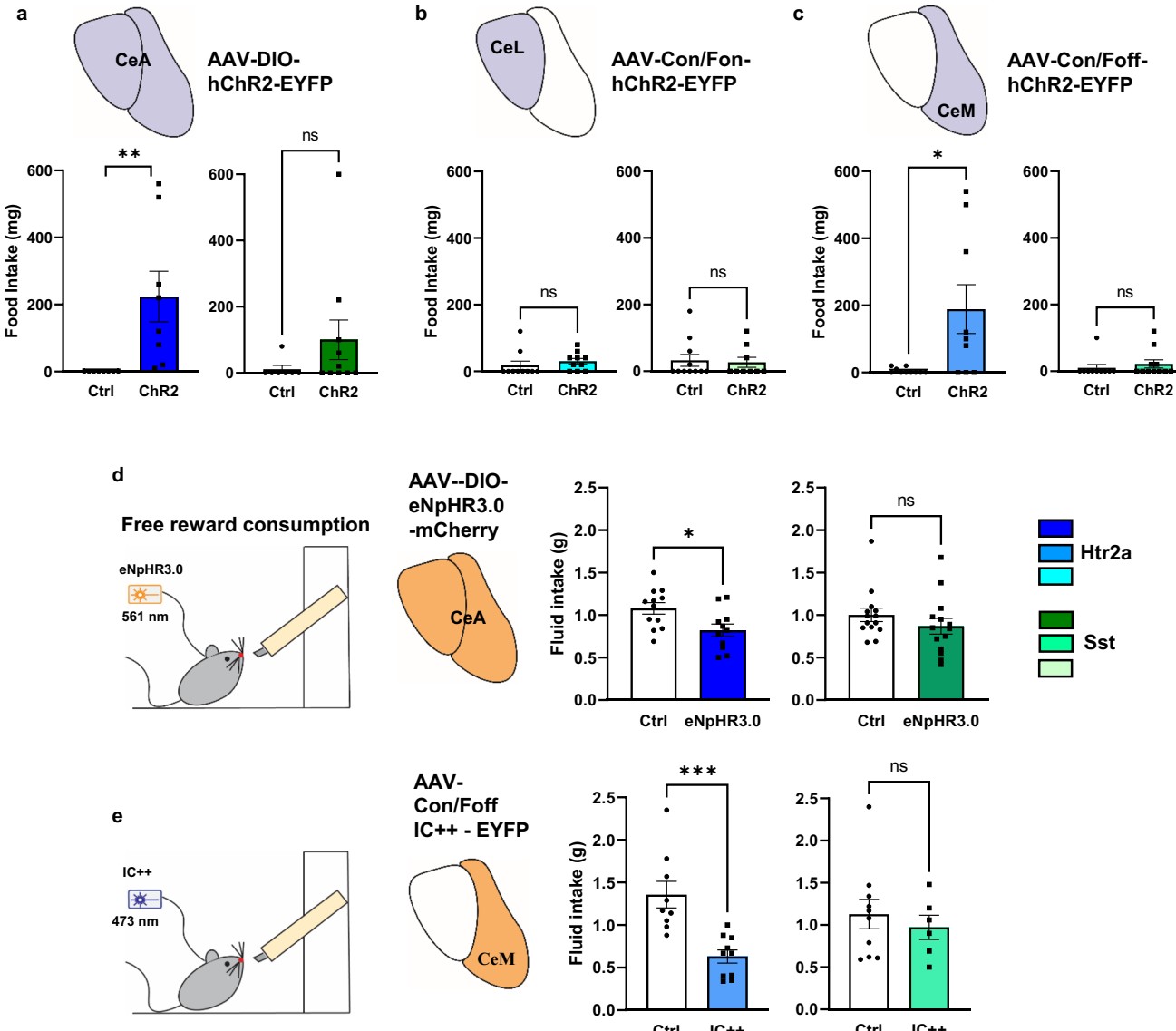

**Fig. 4 | Distinct roles of CeM^Htr2a and CeM^Sst neurons in feeding behavior.**
**a** Photostimulation of the entire population of CeA^Htr2a neurons promoted feeding of satiated mice (two-tailed Wilcoxon signed-rank test $p = 0.0078$). No statistically significant effect after photostimulation CeA^Sst neurons compared to controls (Mann-Whitney two-tailed U test $p = 0.1821$, U = 22). (CeA^Htr2a: n = 8 Ctrl, n = 8 ChR2. CeA^Sst: n = 7 Ctrl, n = 10 ChR2 mice). **b** Photostimulation of CeL^Htr2a or CeL^Sst subpopulations did not promote feeding behavior (CeL^Htr2a: Mann-Whitney two-tailed U test $p = 0.1073$, U = 30.50. CeL^Sst: $p = 0.9183$, U = 48). (CeL^Htr2a: n = 10 Ctrl, n = 10 ChR2. CeL^Sst: n = 11 Ctrl, n = 9 ChR2 mice). **c** Photostimulation of CeM^Htr2a, but not CeM^Sst neurons, increased food intake (CeM^Htr2a: Mann-Whitney two-tailed U test $p = 0.0235$, U = 19.50. CeM^Sst: $p = 0.2783$, U = 41.50). (CeM^Htr2a: n = 10 Ctrl, n = 9

ChR2. CeM^Sst: n = 10 Ctrl, n = 11 ChR2 mice). **d** Photoinhibition of the entire population of CeA^Htr2a neurons in fed mice decreased the ingestion of a palatable liquid reward compared to controls (unpaired two-tailed t test $p = 0.0170$, t = 2.593). No effect by photoinhibition of the entire population of CeA^Sst neurons (Mann-Whitney two-tailed U test $p = 0.1285$, U = 37). (CeA^Htr2a: n = 12 Ctrl, n = 11 eNpHR3.0. CeA^Sst: n = 14 Ctrl, n = 14 eNpHR3.0 mice). **e** Photoinhibition of CeM^Htr2a, but not CeM^Sst, neurons decreased the consumption of a palatable liquid reward compared to controls (CeM^Htr2a unpaired two-tailed t test $p = 0.0005$, t = 4.313; CeM^Sst unpaired two-tailed t test $p = 0.5433$, t = 0.6230). (CeM^Htr2a: n = 9 Ctrl, n = 10 IC ++. CeM^Sst: n = 10 Ctrl, n = 6 IC ++ mice). Values = Mean ± SEM.

analyze how well the neural embeddings decoded specific behavioral features, we applied the RF classifier. The embeddings derived from drinking behavior were used for predicting feeding and those from feeding for predicting drinking. Furthermore, the drinking embeddings were applied to predict Fresubin consumption and vice versa. The results revealed that the neural embeddings decoded the behavioral predictions with higher accuracy compared to the results obtained with shuffled data (Fig. 6i, j, Supplementary Fig. 6f).

Through correlation analysis between behavior and neuronal activity, we found ensembles of CeM^Htr2a and CeM^Sst populations whose activities correlated positively and negatively with reward consumption (Supplementary Fig. 6g). The CeM^Sst populations

responded rather homogeneously to stimulus exposure, with 46–48% of cells displaying positive correlation with all three rewards (Supplementary Fig. 6h). The responses of CeM^Htr2a neurons were more heterogenous, with 39–59% showing positive correlation (Supplementary Fig. 6h). The highest positive correlation of CeM^Htr2a neurons was with feeding (59%), consistent with their observed function in food consumption.

Next, we asked how the activity of individual cells changed between the rewards having different physical properties. We compared all cells that were active during two behavioral sessions, e.g., drinking and feeding or feeding and Fresubin, and asked which fractions of cells kept their correlated activity constant

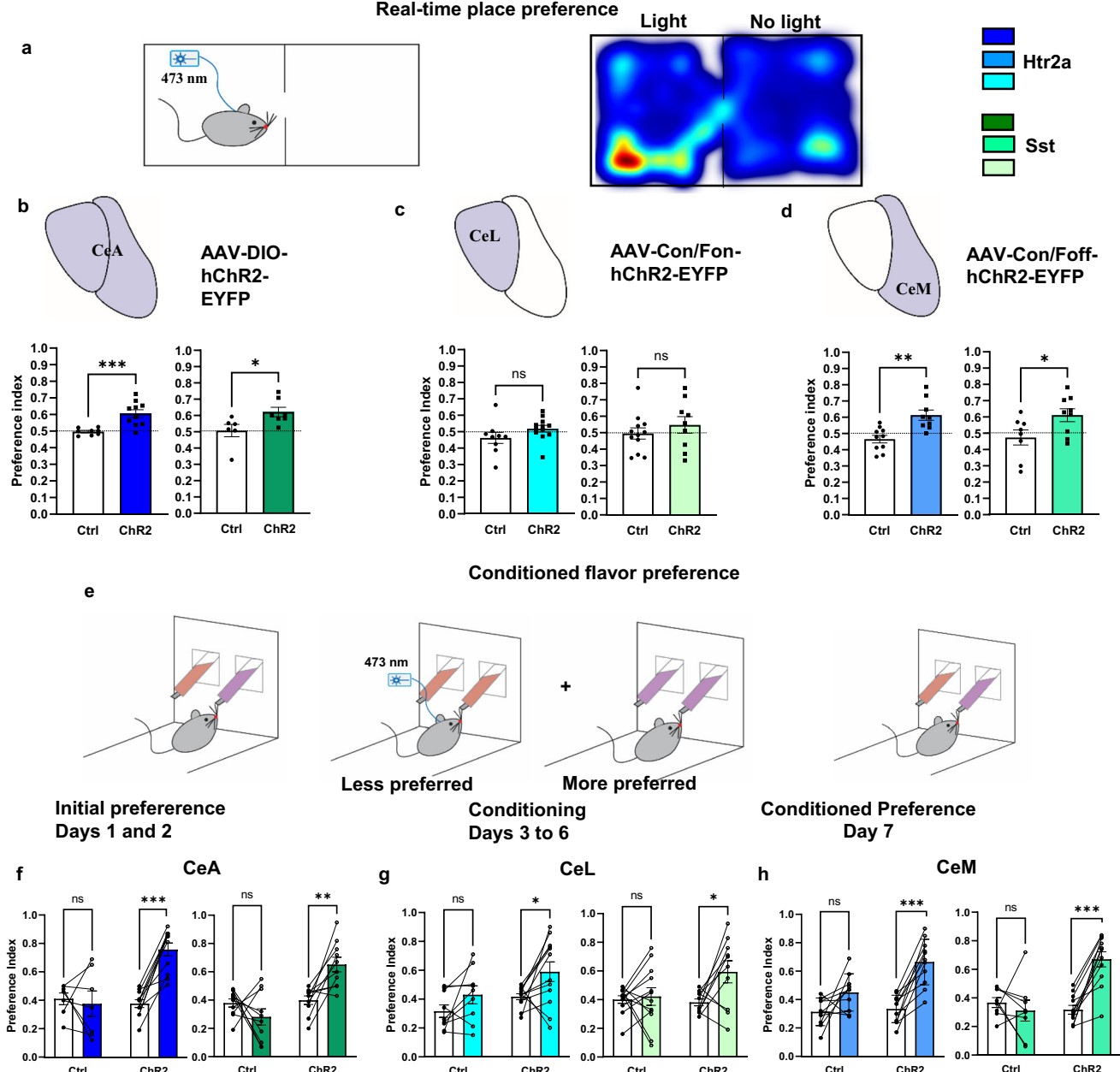

**Fig. 5 | Activation of Htr2a and Sst neurons has a rewarding effect. a** Real-time place preference paradigm. Representative heat map (generated by Ethovision) of a photostimulated CeM^Htr2a::ChR2 mouse in the RTPP task. **b** Preference for the light-paired chamber when the entire populations of CeA^Htr2a or CeA^Sst neurons are activated in the RTPP task, compared to controls (CeA^Htr2a: unpaired two-tailed t test $p = 0.0010$, t = 4.018. CeA^Sst: unpaired two-tailed t test $p = 0.0347$, t = 2.409). (CeA^Htr2a: n = 8 Ctrl, n = 10 ChR2. CeA^Sst: n = 6 Ctrl, n = 7 ChR2 mice). **c** No preference for the light-paired chamber when only CeL^Htr2a or CeL^Sst neurons are activated (Htr2a: unpaired two-tailed t test $p = 0.1521$, t = 1.492, Sst: unpaired two-tailed t test $p = 0.3857$, t = 0.8879). (CeL^Htr2a: n = 9 Ctrl, n = 12 ChR2. CeL^Sst: n = 12 Ctrl, n = 9 ChR2 mice). **d** Preference for the light-paired chamber when only CeM^Htr2a or CeM^Sst neurons are activated (Htr2a: unpaired two-tailed t test $p = 0.0013$, t = 3.844, Sst: unpaired two-tailed t test $p = 0.0383$, t = 2.271). (CeM^Htr2a: n = 10 Ctrl, n = 9 ChR2. CeM^Sst: n = 8 Ctrl, n = 9 ChR2 mice). **e** Conditioned flavor preference test. **f** Conditioned flavor preference index of mice in which the entire populations of CeA^Htr2a (blue) or CeA^Sst (green) neurons were photostimulated in comparison to

control mice (CeA^Htr2a: main effect ChR2, Two-way ANOVA, $F(1,16) = 10.12$, $p = 0.0058$; Bonferroni post-hoc test $p < 0.0001$. CeA^Sst: main effect ChR2, Two-way ANOVA, $F(1,18) = 32.06$, $p < 0.0001$; Bonferroni post-hoc test $p = 0.0057$). (CeA^Htr2a: n = 7 Ctrl, n = 11 ChR2. CeA^Sst: n = 10 Ctrl, n = 10 ChR2 mice). **g** Conditioned flavor preference index of mice in which only the CeL^Htr2a (blue) or CeL^Sst (green) neurons were photostimulated (CeL^Htr2a: main effect ChR2, Two-way ANOVA, $F(1,19) = 5.611$, $p = 0.0286$; Bonferroni post-hoc test $p = 0.0313$. CeL^Sst: main effect Time, Two-way ANOVA, $F(1,20) = 4.918$, $p = 0.0383$; Bonferroni post-hoc test $p = 0.0255$). (CeL^Htr2a: n = 9 Ctrl, n = 12 ChR2. CeL^Sst: n = 12 Ctrl, n = 10 ChR2 mice). **h** Conditioned flavor preference index of mice in which only the CeM^Htr2a (blue) or CeM^Sst (green) neurons were photostimulated (CeM^Htr2a: main effect ChR2, Two-way ANOVA, $F(1,19) = 10.05$, $p = 0.0050$; Bonferroni post-hoc test $p < 0.0001$. CeM^Sst: main effect ChR2: Two-way ANOVA, $F(1,17) = 8.165$, $p = 0.0109$; Bonferroni post-hoc test $p < 0.0001$). (CeM^Htr2a: n = 10 Ctrl, n = 11 ChR2. CeM^Sst: n = 8 Ctrl, n = 11 ChR2 mice). Values = Mean ± SEM.

(positive–positive, or negative–negative correlation) versus fractions that changed their activity (Fig. 6k,l; Supplementary Fig. 6j). When comparing drinking and consuming solid food, the fraction of CeM^Htr2a cells displaying a stable correlation (pos–pos, neg–neg) was

comparable to the fraction of cells that changed their activity (47% versus 53%) (Fig. 6k). Instead, the fraction of CeM^Sst cells that changed their activity was significantly larger than the fraction of cells displaying a stable correlation (63% versus 37%) (Fig. 6k). When

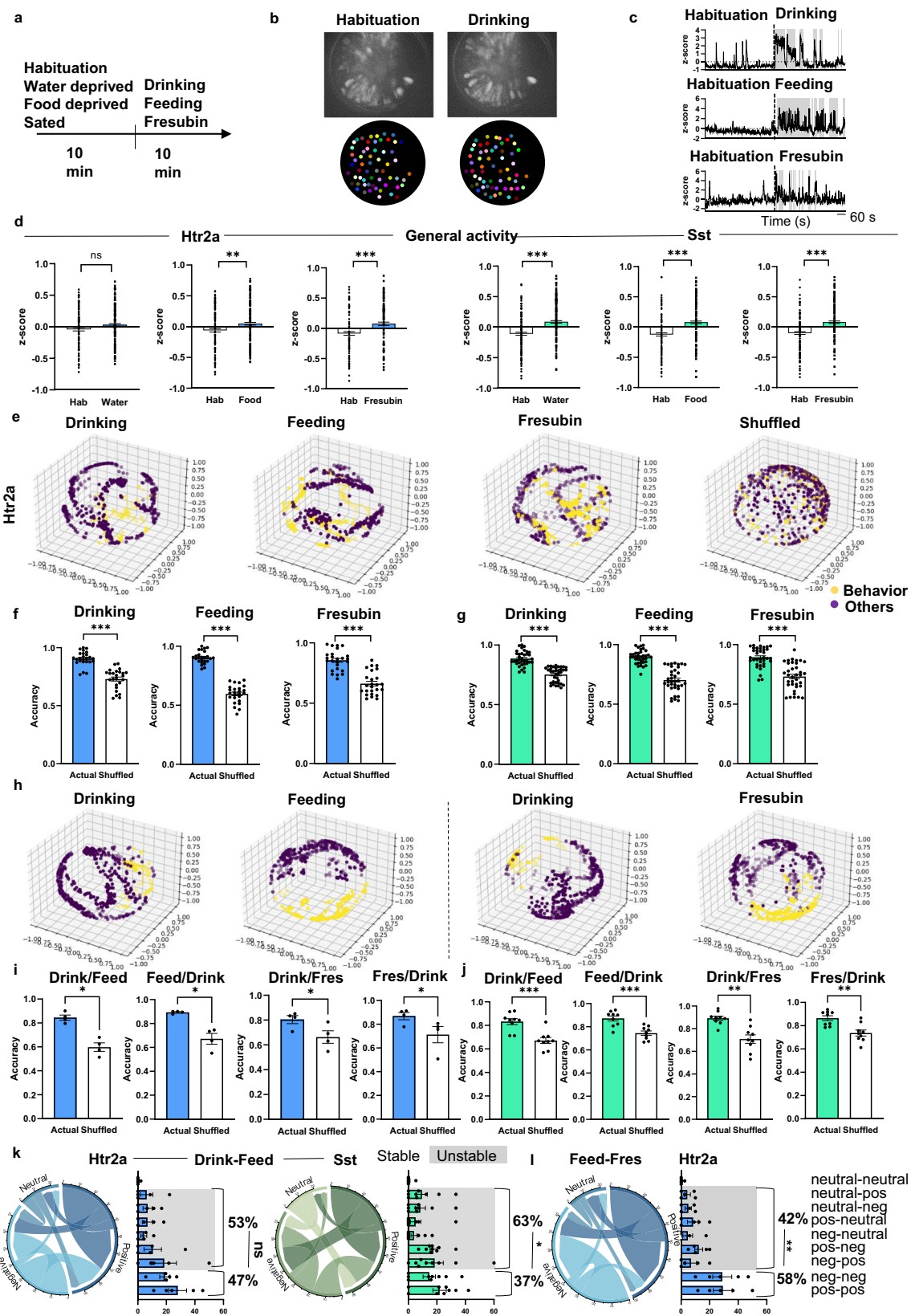

comparing solid food and liquid Fresubin consumption, the fraction of CeM[Htr2a] cells displaying a stable correlation was significantly larger than the fraction of cells that changed their activity (58% versus 42%) (Fig. 6l). Instead, the fraction of CeM[Sst] cells showing a stable correlation was comparable to the fractions of cells that changed their activity (Supplementary Fig. 6j). In summary, at the population level, the majority of CeM[Htr2a] and CeM[Sst] neurons increased their activities during reward consumption and their activity dynamics contributed to and decoded specific behavioral features. Depending on the physical attributes of the rewards, CeM[Htr2a] and CeM[Sst] neurons were recruited into different ensembles with constant or variable correlated activity.

**Fig. 6 | CeM^Htr2a and CeM^Sst neurons are differentially activated by multiple rewarding stimuli. a** Behavioral paradigm. **b** Maximum projection images of calcium responses from CeM^Htr2a neurons during habituation and drinking. **c** Representative traces from CeM^Htr2a neurons during reward consumption. Consumption bouts in gray. **d** Average z-score comparisons of the general activity of CeM^Htr2a and CeM^Sst neurons during habituation and reward (Mann-Whitney two-tailed U test. CeM^Htr2a: water, $p = 0.1771$, $U = 7322$, $n = 112–145$; food, $p = 0.0098$, $U = 9440$, $n = 135–169$; Fresubin, $p = 0.0001$, $U = 4901$, $n = 111–124$. CeM^Sst: water, $p < 0.0001$, $U = 8130$, $n = 143–176$; food, $p < 0.0001$, $U = 7320$, $n = 128–176$; Fresubin $p < 0.0001$, $U = 8119$, $n = 138–168$). **e** 3D Visualization of CEBRA-generated embeddings of CeM^Htr2a neurons considering consumption behavior and neural dynamics over time. Shuffled: drinking data. The three dimensions represent the three CEBRA principle components. Each dot represents a time point. Yellow and purple dots mark consumption and resting times, respectively. Comparison of behavioral decoding accuracy using a Random Forest (RF) algorithm across animals for CeM^Htr2a (**f**) and CeM^Sst neurons (**g**) (drinking, feeding, Fresubin: two-tailed Wilcoxon matched-pairs signed-rank test $p < 0.0001$). (Prediction scores of accuracy of behavior compared to shuffled data: Actual-Shuffled Htr2a, $n = 25$; Sst, $n = 36$). **h** CEBRA-generated embeddings of CeM^Htr2a neurons that were longitudinally detected during drinking and feeding, and during drinking and Fresubin consumption. Accuracy of behavioral decoding through RF, employing CEBRA embeddings of CeM^Htr2a (**i**) and CeM^Sst (**j**) neurons longitudinally detected during two behaviors (paired two-tailed t test. CeM^Htr2a: drinking-feeding: $p = 0.0158$, $t = 4.956$; feeding-drinking: $p = 0.0175$, $t = 4.768$; drinking-Fresubin: $p = 0.0292$, $t = 3.935$; Fresubin-drinking: $p = 0.0337$, $t = 3.724$. CeM^Sst: drinking-feeding: $p = 0.0003$, $t = 6.068$; feeding-drinking: $p = 0.0007$, $t = 5.372$; drinking-Fresubin: $p = 0.0011$, $t = 4.955$; Fresubin-drinking: $p = 0.0012$, $t = 4.867$). (Prediction scores of accuracy of behavior compared to shuffled data: Actual-Shuffled Htr2a, $n = 4$; Sst, $n = 9$). **k** Chord diagrams and bar graphs depicting stable or unstable (gray) correlations of the same CeM^Htr2a and CeM^Sst neurons detected during drinking and feeding (Htr2a: stable v.s. unstable: unpaired two-tailed t test $p = 0.5418$, $t = 0.6371$, $n = 5$ mice. Sst: $p = 0.0193$, $t = 2.602$, $n = 9$ mice). **l** Similar analysis for CeM^Htr2a neurons detected during feeding and Fresubin sessions (stable v.s. unstable: unpaired t test $p = 0.0017$, $t = 4.627$, $n = 5$ mice). Values = Mean ± SEM.

## CeM^Htr2a and CeM^Sst neurons respond differently to stimuli of opposite valence

To investigate what fractions of CeM^Htr2a and CeM^Sst neurons respond specifically to one class of stimuli ("specializers") or exhibit a general response to a broad range of stimuli even of opposite valence ("generalizers"), we exposed the same cohorts of mice expressing GCaMP6m (Supplementary Fig. 7a) to three different aqueous solutions: water, saccharin, and the bitter tastant quinine. During each 10-min session, the stimulus was orally administered to water-deprived mice at minutes 1 and 6 by an experimenter who the mice were well accustomed to. During the remaining time, mice were allowed to freely behave (Fig. 7a). We recorded from an average of 107 CeM^Htr2a and 101 CeM^Sst neurons and an average of 71 neurons was longitudinally detected in different sessions (Supplementary Fig. 7b, c). Many CeM^Htr2a and CeM^Sst neurons showed an increase in calcium responses to all three stimuli during the stimulus exposure periods (Fig. 7b). When analyzing the average population activity across multiple 10-min sessions, we found that the switch from water to saccharin, and water to quinine-added water, did not significantly change activity (although the latter was close to significance). Interestingly, we observed a significant decrease in population activity when switching from saccharin to quinine (Fig. 7c). These results suggest that the switch from sweet to bitter taste quenched the activities of both subpopulations. We applied the CEBRA Hybrid model on the recorded neuronal activities of both CeM^Htr2a and CeM^Sst neurons during the water, saccharin, and quinine behaviors. The observable structures suggested that these behaviors contributed significantly to the alterations in neuronal activity (Fig. 7d). The embeddings of both neuron populations could accurately decode positive and negative stimuli with higher accuracies (97–100%) and lower OOB errors compared to the shuffled data (71–73%) (Fig. 7e; Supplementary Fig. 7e).

Next, we compared the activity during the stimulus exposure episodes with the times in-between and asked which cells were positively or negatively correlated, or showed an activity that was not significantly correlated with stimulus presentation (neutral). We then asked how many cells could be qualified as generalizers showing a stable correlation (positive–positive, negative–negative) between oppositely valenced stimuli, and how many cells would be specializers that switched their correlation. For CeM^Htr2a neurons, during the switch from saccharin to quinine, the fraction of specializers was significantly larger than the fraction of generalizers (60% versus 40%). Conversely, for CeM^Sst cells the pattern was opposite, with generalizers outnumbering specializers (64% versus 36%) (Fig. 7f–,). Other comparisons did not yield significant differences between generalizers and specializers (Supplementary Fig. 7f). In summary, these results suggest that the activities of CeM^Htr2a and CeM^Sst neurons contribute to the detection of stimuli of opposite valence, and that the CeM^Htr2a population contains more cells that specialize in encoding valence-specific stimuli than CeM^Sst neurons.

To disentangle between activation of a fluid intake motor program (as suggested by the photoactivation experiments) and valence detection, we next asked if photoactivation of CeM^Htr2a or CeM^Sst neurons would be sufficient to induce consumption of quinine adulterated water. Neurons were photoactivated during 30 min in which water-deprived mice were exposed to a 10 mM quinine solution. While photoactivated control mice avoided the bitter solution, activation of CeM^Htr2a and CeM^Sst neurons stimulated quinine consumption (Fig. 7i). The same mice were tested with a subtler quinine solution (100 μM) under similar conditions and were found to consume more fluid compared to controls (Fig. 7j). Comparing the intake of the two quinine concentrations by the same mice, we found that ChR2-expressing mice drank significantly more of the lower-concentrated solution. These findings suggest that photoactivation of CeM^Htr2a and CeM^Sst neurons stimulates a fluid intake motor program, but that the mice are still able to distinguish varying concentrations of unpleasant taste.

## Appetitive CeM neurons are inhibited by anorexigenic CeA^PKCδ neurons

Next, we asked how the activities of appetitive CeM^Htr2a and CeM^Sst may be regulated. Previous studies had suggested that putative appetitive CeA neurons may be under inhibitory control of anorexigenic CeA^PKCδ neurons[6] and that CeA^Htr2a neurons engage in reciprocal inhibitory connections with CeA^PKCδ neurons[5]. This raised the possibility that CeM^Htr2a and CeM^Sst neurons may also receive direct inhibitory input from CeA^PKCδ neurons that reside in the CeL/C subregion. We hypothesized that inhibition of CeA^PKCδ neurons would disinhibit and thereby activate appetitive CeM^Htr2a and CeM^Sst neurons, whereas activation of CeA^PKCδ neurons would inhibit them. We first confirmed that photoactivation of PKCδ cells suppressed water consumption compared to photostimulated control mice (Fig. 8a–d). In a 10-min ON/OFF stimulation protocol (similar to Fig. 2, but starting with Light ON), the control group consumed most of the water during the initial 10 min, then gradually decreased consumption as the mice became satiated, regardless of the light phase. In contrast, CeA^PKCδ neurons consumed water only during the Light OFF phases (Fig. 8d). Conversely, photoinhibition of CeA^PKCδ neurons using Cre-dependent Halorhodopsin (Fig. 8e, f) resulted in increased water intake compared to photostimulated control mice (Fig. 8g). Notably, anxiety did not appear to affect the behavioral outcomes, as the mice subjected to photoactivation or inhibition of CeA^PKCδ neurons spent a comparable amount of time in the center-zone during the open-field test as the control group (Fig. 8h, i).

To demonstrate a functional connection between CeA^PKCδ and the appetitive CeM subpopulations, we generated the mouse line PKCδ-

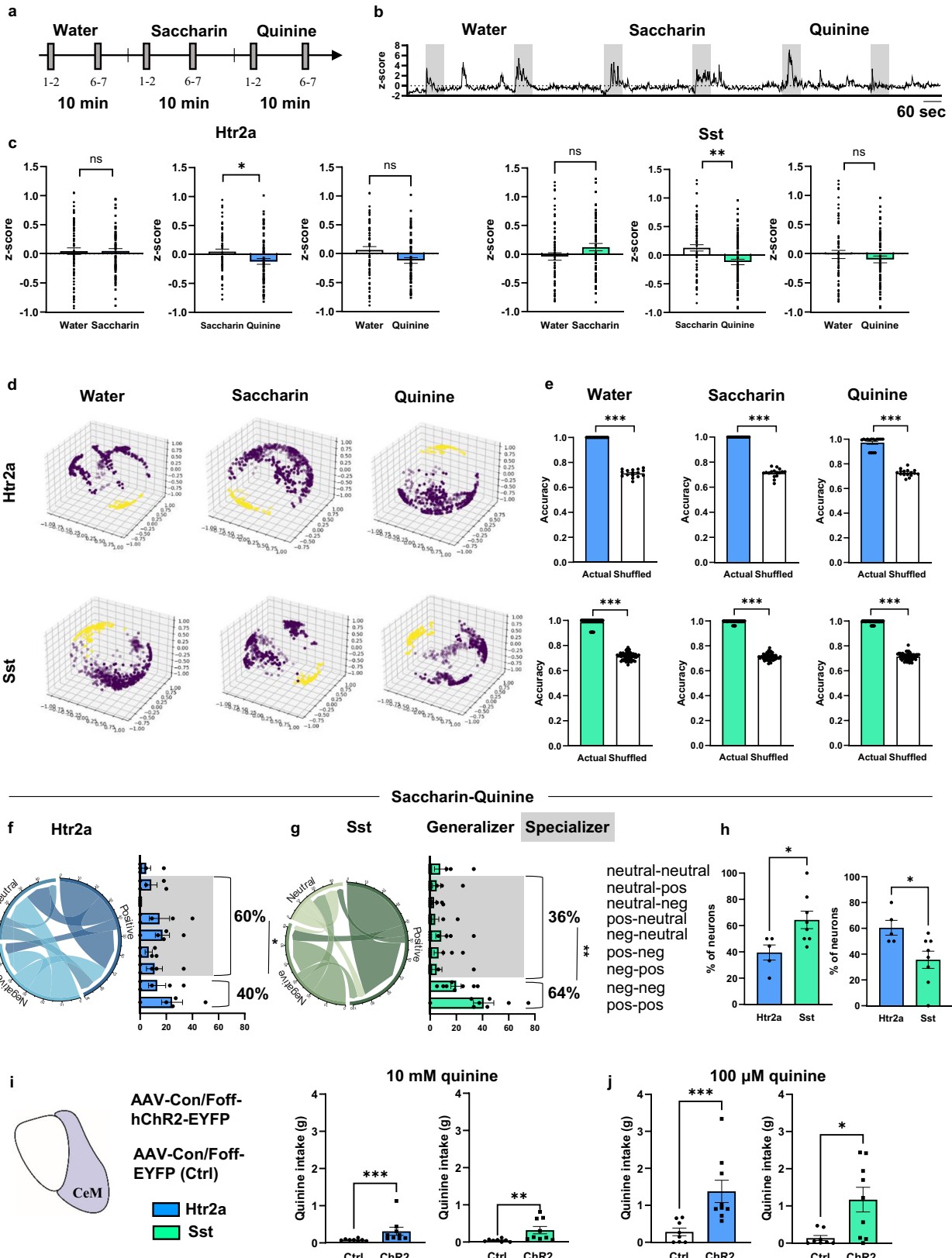

Flp, that expresses the Flp recombinase specifically under the control of the PKCδ promoter (Fig. 8j). We validated the expression of the Flp recombinase by crossing the PKCδ-Flp mouse line with a Flp-dependent reporter mouse and quantified the numbers of reporter-positive cells versus PKCδ immunostaining. The results revealed that 89% of the PKCδ immunopositive cells colocalized with the reporter

(Fig. 8j). Using an intersectional approach, we crossed PKCδ-Flp mice with Htr2a-Cre mice also carrying a Cre-dependent tdTomato reporter (Ai9) to later visualize CeM[Htr2a] neurons in slices. Similar crosses were done with Sst-Cre mice (Fig. 8k). We then injected a Flp-dependent ChR2-EYFP AAV into the CeA to photoactivate CeA[PKCδ] neurons in slices. With this design, it was possible to patch and record from CeM

**Fig. 7 | Different salient stimuli elicit responses in CeM^Htr2a and CeM^Sst neurons. a** Behavioral paradigm. **b** Representative trace from a CeM^Htr2a neuron showing increased neuronal activity during the 1-min consumption bouts (gray color). **c** Average z-score comparisons of the activities of the same CeM^Htr2a and CeM^Sst neurons recorded in the three different conditions (two-tailed Wilcoxon matched-pairs signed-rank test. CeM^Htr2a: water-saccharin, $p = 0.9477$, n = 80; saccharin-quinine, $p = 0.0175$, n = 71; water-quinine, $p = 0.0714$, n = 67. CeM^Sst: water-saccharin, $p = 0.1836$, n = 69; water-quinine, $p = 0.6808$, n = 63; saccharin-quinine, paired two-tailed t test $p = 0.0067$, t = 2.788, n = 78). **d** CEBRA-generated embeddings of CeM^Htr2a and CeM^Sst neurons in the three conditions. **e** Comparison of behavioral decoding accuracy using RF across animals for Htr2a (blue) and Sst (green) neurons (water, saccharin, quinine: two-tailed Wilcoxon matched-pairs signed-rank test $p < 0.0001$). (Prediction scores of accuracy of behavior compared to shuffled data. Htr2a: Actual-Shuffled n = 16. Sst: Actual-Shuffled: n = 57). Chord diagrams and bar graphs showing the stable ("generalizers") or unstable correlation ("specializers") of the same CeM^Htr2a (**f**) and CeM^Sst neurons (**g**) when switching from saccharin to quinine. Specializers in gray (CeM^Htr2a unpaired two-tailed t test $p = 0.0332$, t = 2.569, n = 5 mice; CeM^Sst $p = 0.0087$, t = 3.047, n = 8 mice). **h** Percentages of generalizers (left) and specializers (right) within the CeM^Htr2a and CeM^Sst populations (Generalizer: unpaired two-tailed t test $p = 0.0259$, t = 2.573. Specializer: $p = 0.0261$, t = 2.570). (Htr2a: n = 5, Sst: n = 8). **i** Photoactivation of CeM^Htr2a (blue) and CeM^Sst neurons (green) promotes 10 mM quinine intake compared to controls (CeM^Htr2a: Mann-Whitney two-tailed test p = 0.0007, U = 3.500. CeM^Sst: p = 0.0016, U = 5). (CeM^Htr2a: n = 8 Ctrl, n = 9 ChR2; CeM^Sst: n = 8 Ctrl, n = 9 ChR2 mice). **j** Same as in (**i**), except for 100 μM quinine (CeM^Htr2a: Mann-Whitney two-tailed test p = 0.0003, U = 2. CeM^Sst: p = 0.0151, U = 11). Group sizes as in (**i**). CeM^Htr2a::ChR2 and CeM^Sst::ChR2 mice drink larger amounts of 100 μM compared to 10 mM quinine (CeM^Htr2a main effect quinine: Two-way ANOVA, $F_{(1,15)} = 14.83$, $p = 0.0016$; multiple comparison $p = 0.0003$. CeM^Sst main effect quinine: Two-way ANOVA, $F_{(1,15)} = 8.711$, $p = 0.0099$; multiple comparison $p = 0.0015$). (n = 8 Ctrl, n = 9 ChR2 mice). Values = Mean ± SEM.

tdTomato-positive neurons while photoactivating CeA^PKCδ neurons in the CeL (Fig. 8k). These results showed that photoactivation of CeA^PKCδ neurons suppressed current-induced firing of cells in the CeM (Fig. 8l, m). Additionally, we could record inhibitory postsynaptic currents from CeM^Htr2a and CeM^Sst neurons, suggesting evidence for a monosynaptic connection from CeA^PKCδ to both CeM subpopulations (Fig. 8n, o). Together, these findings demonstrate that CeA^PKCδ neurons suppress water intake and inhibit the activities of CeM^Htr2a and CeM^Sst neurons in slices, consistent with a model in which the activities of CeM^Htr2a and CeM^Sst neurons are under inhibitory control of CeA^PKCδ neurons in vivo.

### Htr2a and Sst neurons send projections to brain regions associated with reward processing

After having identified one of the possible mechanisms regulating the activity of Htr2a and Sst neurons in the CeM, we proceeded to anatomically map the long-range outputs of these neurons. To identify the major output targets of Htr2a and Sst neurons in the CeL and CeM, we selectively expressed either Con/Fon or Con/Foff YFP viruses in Htr2a/Sst-Cre::Wfs1-FlpoER mice. To determine the relative strength of the projections, we calculated the integrated fluorescence intensities in the output region normalized to background and injection sites. Our analysis revealed that for Htr2a neurons, the CeM fraction projected to a larger number of outputs compared to the CeL fraction (19 versus 8), whereas for Sst neurons, CeM and CeL fractions projected to similar numbers of outputs (20 versus 16) (Supplementary Figs. 8 and 9). A number of brain regions received strong projections from both Htr2a and Sst CeL and CeM fractions, including the bed nucleus of the stria terminalis (BNST), the lateral vestibular nucleus (LAV), the lateral and medial parabrachial nucleus (LPBN, MPBN), and the midbrain reticular nucleus (MRN) (Fig. 9a–d). The interstitial nucleus of the posterior limb of the anterior commissure (IPAC) and the substantia innominata (SI) received stronger projections from the CeM compared to the CeL fractions. Few brain regions received enriched projections from CeM^Sst neurons, including the lateral and medial geniculate complex (LG, MG) and the ventral posterolateral/medial nucleus of the thalamus (VPL/M) (Fig. 9a–d; Supplementary Figs. 8 and 9). Several of these regions are known to be involved in rewarding and consummatory behaviors.

### The appetitive functions of CeM^Htr2a and CeM^Sst neurons are mediated by projections to the PBN

Based on the known functions of the PBN serving as a hub for sensory information relevant to food and water intake, receiving inputs from interoceptive and exteroceptive sources[38–44], and driving aversive emotional behaviors, we hypothesized that inhibition of PBN neurons by CeM^Htr2a or CeM^Sst neurons may promote drinking and/or feeding behavior. To explore the functions of these neuronal projections, we injected a Con/Foff ChR2 virus, or similar control virus, bilaterally into

the CeA of Htr2a-Cre::Wfs1-FlpoER or Sst-Cre::Wfs1-FlpoER mice and placed optic fibers bilaterally above the PBN (Fig. 9e, Supplementary Fig. 10a, b). Photoactivation of the presynaptic terminals in the PBN of CeM^Htr2a or CeM^Sst neurons led to increased drinking in both water-deprived and hydrated mice during both a 30 min and a 10 min Light ON/OFF stimulation protocol (Fig. 9f, g, Supplementary Fig. 10c–h.). Furthermore, photoactivation of the CeM^Htr2a → PBN projectors stimulated feeding behavior, whereas, as expected, no effect was observed for the CeM^Sst → PBN projectors (Fig. 9h, i). Moreover, both CeM^Htr2a → PBN and CeM^Sst → PBN projectors promoted rewarding behavior in real-time place preference (Fig. 9j, k) and conditioned flavor preference assays (Supplementary Fig. 10i–l). No changes in locomotion or anxiety were observed in the open-field test (Supplementary Fig. 10m, n). These findings suggest that CeM^Htr2a and CeM^Sst neurons promote appetitive and reward behavior through inhibition of PBN neurons.

## Discussion

In this report, we have used an intersectional genetics approach to independently target four putative appetitive neuron subpopulations in the central amygdala: two in the CeL and two in the CeM. We found that neurons mediating water and/or food consumption are confined to the CeM. Separate CeM subpopulations exist for water only (CeM^Sst), and water or food consumption (CeM^Htr2a). All four subpopulations are intrinsically positively reinforcing in a conditioned flavor preference assay. Through in vivo calcium imaging we observed that the majority of CeM^Htr2a and CeM^Sst neurons increased their activity during reward consumption and that their activity dynamics contributed to and decoded specific behavioral features. Depending on the type of rewards, CeM^Htr2a and CeM^Sst neurons were recruited into different ensembles with constant or variable correlated activity. Calcium imaging further suggests that the activities of CeM^Htr2a and CeM^Sst neurons contribute to the detection of stimuli of opposite valence, and that the CeM^Htr2a population contains more cells that specialize in encoding valence-specific stimuli than CeM^Sst neurons. At the microcircuit level, the activity of appetitive CeM neurons is controlled by inhibitory signals from CeA^PKCδ neurons and, in turn, appetitive CeM neurons form long-range inhibitory projections to the PBN to promote appetitive and reward behavior (Fig. 9l). In summary, this study provides a comprehensive functional characterization of molecularly- and anatomically-defined CeA neurons and their roles in appetitive behaviors.

Intersectional genetics using two recombinases has become a valuable tool to address specific neuron subtypes within a larger population of neurons, for example, distinct subtypes of midbrain dopaminergic neurons or hypothalamic POMC neurons[45,46]. This method has also been useful to manipulate specific neurons within an anatomically well-defined area, for example, specific Cre+ neuron

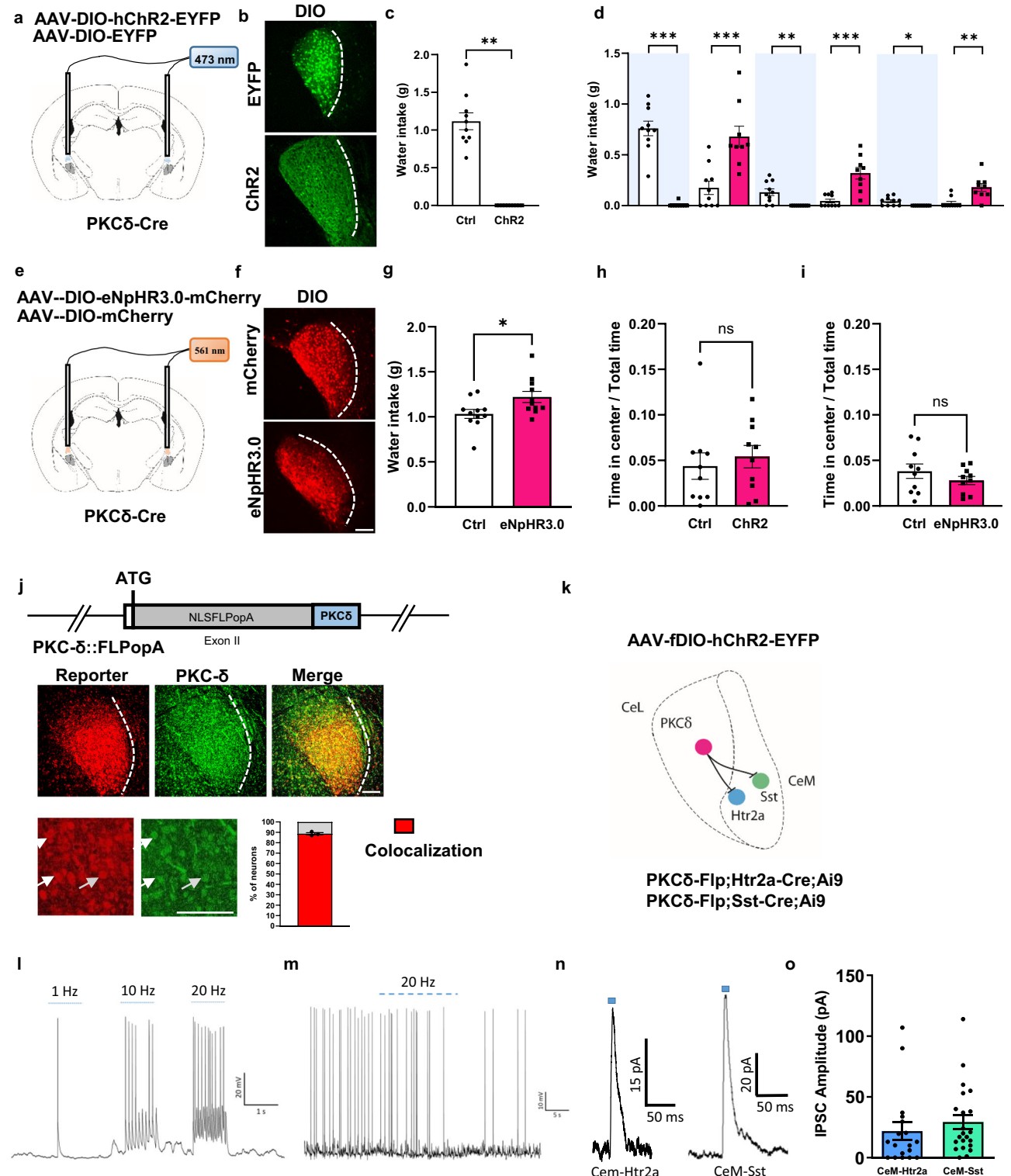

populations in the spinal cord, and to assess the functional consequences without interference by Cre recombination in the brain[47]. Here, we have used the Wfs1-FlpoER mouse line that expresses the optimized and tamoxifen-inducible FlpoER recombinase in the CeL subdivision of the CeA, to a much higher degree (17-fold) than in cells of the CeM. In combination with a specific Cre line and a Con/Fon Boolean reporter virus, we could target between 4 and 10 times more Cre+ cells in the CeL than CeM. Conversely, with a Con/Foff virus, we could target between 2.3 and 2.7 times more Cre+ cells in the CeM than CeL. These results indicate that the system is effective at producing an enrichment of CeL versus CeM subpopulations. The inferior performance of the Con/Foff virus may have resulted from incomplete coverage of Wfs1-Flp expression in CeL neurons. Cre-positive CeL cells that do not co-express Flp would express the Con/Foff reporter and thereby lower the CeM:CeL ratio. As for most biological systems, the expression in the CeL versus CeM subpopulations is not black-and-white. The strength of the system relies on the combination of the INTRSECT viruses with photoactivation and -inhibition experiments.

The CeL is largely composed of three cell populations: CeL^PKCδ neurons that inhibit food or water consumption (this study and[6,7,48–50]),

**Fig. 8 | Inhibition of appetitive CeM neurons by CeA^PKCδ neurons. a** Viruses injected and optic fiber placement in the CeA (modified from Allen Mouse Brain Atlas, mouse.brain-map.org). **b** EYFP and ChR2 expression in the CeL of PKCδ-Cre mice. All mice used for behavior showed similar results. **c** Water intake during 30 min CeA^PKCδ photoactivation in water-deprived mice (two-tailed Wilcoxon signed-rank test $p = 0.0020$). (n = 10 mice each). **d** Water intake during photo-activation of CeA^PKCδ neurons in a 10 min Light ON/OFF behavioral paradigm (Light ON, light blue shading) by water-deprived mice (0–10 min, Mann-Whitney two-tailed U test $p < 0.0001$, U = 0; 10–20 min, $p = 0.0004$, U = 5; 20–30 min, two-tailed Wilcoxon signed-rank test $p = 0.0078$; 30–40 min, Mann-Whitney two-tailed U test $p = 0.0002$, U = 4; 40–50 min two-tailed Wilcoxon signed-rank test $p = 0.0312$; 50–60 min Mann-Whitney U test $p = 0.0011$, U = 9). (n = 10 mice each). **e** Same as in (**a**) except for eNpHR3.0 expression (modified from Allen Mouse Brain Atlas, mouse.brain-map.org). **f** Same as in (**b**) except for mCherry and eNpHR3.0 expression. **g** Water intake during 30 min photoinhibition of CeA^PKCδ neurons by water-deprived mice (unpaired two-tailed t test $p = 0.0282$, t = 2.365). (n = 11 mice

each). Time in the center during an OF test by photoactivated PKCδ-Cre::ChR2 (**h**) (Mann-Whitney two-tailed U test $p = 0.4359$, U = 39, n = 10 mice each) and by photoinhibited PKCδ-Cre::eNpHR3.0 (**i**) (unpaired two-tailed t test $p = 0.2719$, t = 1.133, n = 11 mice each). **j** PKCδ-Flp transgene. NLSFLPo-pA, optimized FLP with NLS and polyA recognition sequence. mCherry (red) and PKCδ (green) expression in the CeA of PKCδ-Flp;FPDi mice. Fraction of mCherry+ among PKCδ cells: $88.7 \pm 1.0\%$ (n = 3 brains, 3 sections per brain). **k** Intersectional strategy to map PKCδ-CeM connectivity. **l** Example trace of a CeA^PKCδ neuron photostimulated with 1, 10, and 20 Hz in slices. **m** Representative example of photostimulated (20 Hz) CeA^PKCδ neuron suppressing current-injection-induced firing of a CeM^Sst neuron. **n** Representative traces of induced inhibitory postsynaptic current (IPSC) in CeM^Htr2a (left) and CeM^Sst (right) cells after photoactivation of CeA^PKCδ neurons. **o** Quantification of IPSC amplitudes of recorded CeM^Htr2a and CeM^Sst neurons (n = 22 for CeM^Htr2a, n = 18 for CeM^Sst neurons). Values = Mean ± SEM. Scale bar: 115 μm.

and Sst+ neurons that can be further subdivided into CeL^Sst and CeL^Nts/Tac2 cells[7,13,22]. Both Sst+ populations in CeL heavily overlap with Htr2a-Cre expressing cells[13]. Here, by targeting Sst-Cre- or Htr2a-Cre-positive cells in the CeL, we obtained no evidence that they could promote water or food consumption. This was a surprising finding in light of earlier reports showing that Sst+, Nts+, Pnoc+, and Htr2a+ neurons promoted water, palatable fluid, and/or palatable food intake[5,7,23,24]. However, these reports did not distinguish between CeL and CeM subpopulations, raising the possibility that it was indeed the CeM subpopulations that drove the behavior. The study by Kim et al.[7], attempted to target the anatomical subregions with stereotaxic viral injections and reported that both CeL and CeM subpopulations of Sst+ and Nts/Tac2+ neurons promoted water intake. This result is contrary to ours, and may in part be due to the different techniques employed. Further experiments would be necessary to solve this discrepancy.

Contrary to the CeL, the Sst+ and Htr2a-Cre+ subpopulations in the CeM are largely separate populations[13]. Here, we show that CeM^Sst neurons promote water, but not solid food intake, whereas CeM^Htr2a neurons promote water and food intake. These findings are consistent with and extend earlier observations: Nts+ neurons (a subpopulation of Sst neurons) promote ethanol and palatable fluid, but not solid food, consumption[24]; photoactivation of Tac2+ or CRH+ cells (sub-populations of Sst neurons) has no positive effect on feeding[6]; Pnoc+ neurons (heavily overlapping with Htr2a-Cre-expressing neurons) promote palatable food consumption[23], photoactivation of NPY neurons, located in the CeM and overlapping with Htr2a neurons, increases food intake[51].

The optogenetic experiments are supported by in vivo calcium imaging data which indicate that the majority of CeM^Htr2a and CeM^Sst neurons increased their activity during consumption of water and food rewards. CEBRA analysis revealed that their activity dynamics contributed to and decoded specific behavioral features. Depending on the type of rewards, CeM^Htr2a and CeM^Sst neurons were recruited into different ensembles whose activities correlated positively or negatively with reward consumption. The highest positive correlation of CeM^Htr2a neurons was with feeding (59%), consistent with their observed function in food consumption. When switching between rewards of different physical attributes (e.g., solid food and liquid Fresubin), CeM^Htr2a neurons more often displayed a stable correlation, while CeM^Sst neurons more often changed their activity. These results are consistent with previously suggested models[11,12] in which CeA^Sst neurons participate in discriminating between stimuli that differ in their sensory/physical properties, such as taste and texture. In contrast to CeM^Sst neurons, CeM^Htr2a neurons may encode a broader range of information, such as the innately affective properties of the stimulus and the animal's internal hunger state. CeM^Htr2a cells are activated by fasting and the hunger hormone ghrelin[13] and here, we found that a higher percentage of CeM^Htr2a cells were positively correlated with

feeding when the animals were hungry compared to when they were fed and consuming Fresubin. Hence, CeM^Htr2a neuron activity is less correlated with the physical attributes of the stimulus, and more with its palatability and rewarding properties.

If the activity of CeM^Sst neurons increases during water licking and feeding bouts, why is ectopic activation of these neurons sufficient to promote drinking, but not food consumption? Previously, it was shown that the activity of Sst+ neurons (in the entire CeA) arose later than the animal's licking responses following water delivery, suggesting that the activity of these neurons did not promote licking[12]. Instead, Sst+ neurons may drive water consumption by conveying stimulus-specific signals to downstream reward centers. Solid food consumption involves additional aspects such as handling and biting the food, and these aspects may only be driven by CeM^Htr2a, but not CeM^Sst neurons. Further work will be necessary to test this hypothesis.

At the population level, Sst+ CeA neurons were previously shown to encode a range of appetitive and aversive stimuli. Many individual Sst+ neurons displayed high selectivity to only one class of stimuli ("specializers"), some responded to multiple stimuli, sometimes of opposite valence ("generalizers")[12]. Similar heterogeneity may be present in other CeA populations, such as Nts+ neurons which promote the intake of palatable fluids, but not solid food[24]. Here, we also recorded the activity of individual CeM neurons when animals experienced a switch between an appetitive and an aversive liquid. For CeM^Htr2a neurons, the fraction of specializers was significantly larger than the fraction of generalizers when switching from saccharin to quinine solution. Interestingly, for the same switch in stimuli, the pattern was opposite for CeM^Sst cells, with generalizers outnumbering specializers. We conclude that the CeM^Htr2a population contains more cells that innately specialize in encoding valence-specific stimuli than the CeM^Sst population. In other words, for CeM^Htr2a neurons that may integrate physical and rewarding properties of the stimulus with the animal's hunger state, the switch from saccharin to quinine represents an important valence switch that causes a change in activity for most neurons. For CeM^Sst cells that mainly respond to stimuli that differ in their sensory/physical properties, a switch between a sweet and a bitter liquid represents a minor difference in physical attributes that did not cause a change in activity for most neurons.

We also showed that photoactivation of CeM^Htr2a or CeM^Sst neurons was sufficient to induce consumption of quinine adulterated water and the amount consumed negatively correlated with the dose of quinine. These findings suggest that photoactivation of CeM^Htr2a and CeM^Sst neurons stimulates a fluid intake motor program, but that the mice are still able to distinguish varying concentrations of unpleasant taste. We conclude that photoactivation of the neurons has two effects that contribute to fluid consumption: First, the activation of a fluid intake motor program that drives drinking independent of internal state and despite strongly aversive taste, and second, the generation of

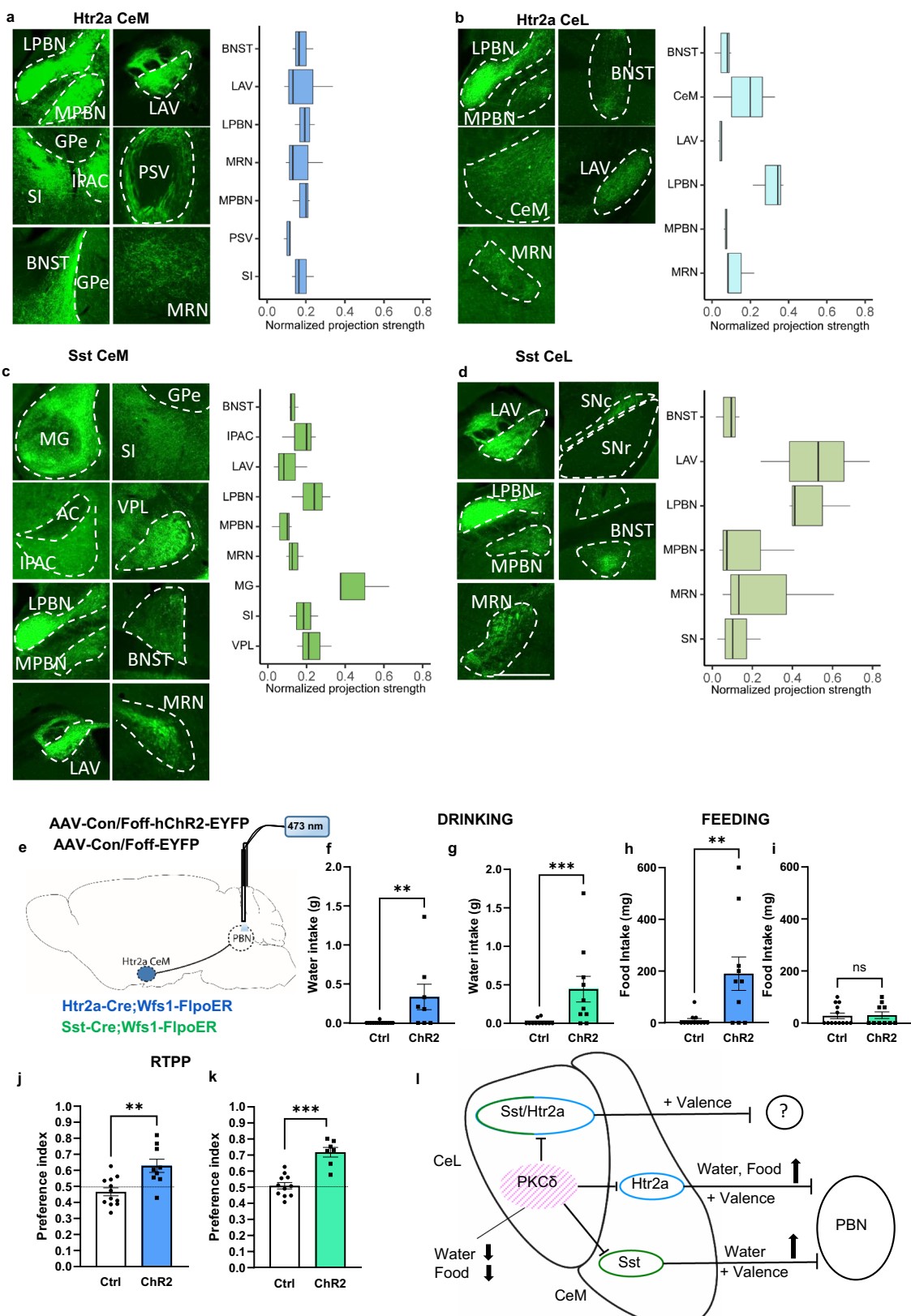

a rewarding effect that counterbalances the bitter taste. The stronger the bitter taste, the more it cancels out the rewarding effects and the less is consumed by the mice.

Previously, the activities of Sst+ and Htr2a-Cre-expressing neurons (of the entire CeA) were shown to be intrinsically rewarding in RTPP assays, and intrinsically reinforcing in intracranial self-

stimulation and reward learning assays[5,7,52]. Here, we show that both CeM subtypes drive RTPP and conditioned flavor preference. It is likely that this activity contributes to the promotion of water or food intake. The CeL subtypes of Sst+ and Htr2a-Cre-expressing neurons also displayed modest reinforcing activity in the conditioned flavor preference assay, but perhaps not in RTPP. These observations are in line

**Fig. 9 | Output regions of Htr2a and Sst neurons and functional analysis of CeM→PBN projectors.** Brain regions that receive projections from CeM[Htr2a] (**a**), CeL[Htr2a] neurons (**b**), CeM[Sst] (**c**), and CeL[Sst] neurons (**d**), and corresponding box-plots with quantification of fluorescence intensities (n = 3 mice per condition). Values = Median (Center) ± Min/Max (whiskers). 25–75 percentile (box). Scale bar: 500 μm. Abbreviations: AC anterior commissure, BNST bed nucleus of the stria terminalis, CeL central amygdala lateral part, CeM central amygdala medial part, GPe globus pallidus, external segment, IPAC interstitial nucleus of the anterior commissure, LAV lateral vestibular nucleus, MG medial geniculate complex, MRN midbrain reticular nucleus, L/MPBN lateral/medial parabrachial nucleus, PSV principal sensory nucleus of the trigeminal, SI Substantia innominata, SN substantia nigra, VPL ventral lateral nucleus of the thalamus. **e** Bilateral CeA viral injection and bilateral optic fiber placement above the PBN (modified from Allen Mouse Brain Atlas, mouse.brain-map.org). Water intake during photoactivation of PBN projections of CeM[Htr2a] (**f**) or CeM[Sst] (**g**) neurons by normally hydrated mice compared to controls (CeM[Htr2a]: Mann-Whitney two-tailed U test p = 0.0021, U = 22.50 and CeM[Sst]: p = 0.0008, U = 13). (CeM[Htr2a] to PBN: n = 14 Ctrl, n = 8 ChR2. CeM[Sst] to PBN: n = 11 Ctrl, n = 10 ChR2 mice). Food intake during photoactivation of PBN projections of CeM[Htr2a] (**h**) or CeM[Sst] (**i**) neurons by satiated mice compared to controls (CeM[Htr2a]: Mann-Whitney two-tailed U test p = 0.0061, U = 23 and CeM[Sst]: p = 0.9265, U = 63.50). (CeM[Htr2a] to PBN: n = 12 Ctrl, n = 10 ChR2. CeM[Sst] to PBN: n = 13 Ctrl, n = 10 ChR2 mice). Preference for the light-paired chamber in the RTPP task when PBN projections of CeM[Htr2a] (**j**) or CeM[Sst] (**k**) neurons are activated, compared to controls (CeM[Htr2a]: unpaired two-tailed t test p = 0.0023, t = 3.522 and CeM[Sst]: p < 0.0001 t = 5.949). (CeM[Htr2a] to PBN: n = 12 Ctrl, n = 9 ChR2. CeM[Sst] to PBN: n = 11 Ctrl, n = 7 ChR2 mice). Values = Mean ± SEM. **l.** The CeA appetitive microcircuit. All CeL and CeM appetitive neurons encode positive valence and form long-range projections. CeM[Htr2a] and CeM[Sst] neurons promote water intake, CeM[Htr2a] neurons promote feeding. They are inhibited by anorexigenic CeA[PKCδ] neurons. Activation of PBN projections of CeM[Htr2a] or CeM[Sst] neurons promotes food and water intake.

with the notion that Sst+ neurons are required for learning and that Sst + neurons projecting to substantia nigra (SN) dopaminergic neurons participate specifically in reward learning[12].

The appetitive inputs that lead to activation of CeM subtypes are not well understood. Brain regions controlling water intake include the anterior cingulate cortex and their projections to the amygdala complex[53] and the peri-locus coeruleus[54]. Food-related inputs may come from insular cortex[21,55], PBN[38,42], the arcuate nucleus[5,56] and the parasubthalamic nucleus[5,57], from fasting, and the hunger hormone ghrelin[13]. Here, we have shown that both CeM subtypes are under inhibitory control of CeA[PKCδ] neurons residing in the CeL. When animals reach satiety during consumption of water or food, the behavior is terminated by activation of CGRP+ PBN neurons[38,41] that project to the CeA and activate PKCδ neurons. Conversely, when animals are thirsty or hungry, CeA[PKCδ] neurons may become inhibited, either through the local CeL inhibitory network, or by long-range inhibitory inputs to CeA[PKCδ] neurons, and this will disinhibit CeM[Htr2a] and CeM[Sst] neurons favoring the expression of appetitive behavior.

A number of brain regions received strong projections from both Htr2a and Sst CeL and CeM fractions. Among them, the BNST plays a central role in reward-related behaviors[58], while also influencing feeding through its projections to the lateral hypothalamus (LH)[59,60]. The parabrachial nucleus is a sensory relay receiving an array of interoceptive and exteroceptive inputs relevant to taste and ingestive behavior, pain, and multiple aspects of autonomic control[38,40,44,61,62]. Brain regions that receive stronger projections from CeM than CeL neurons include the IPAC, located within the extended amygdala, and known to regulate energy homeostasis[63] and the SI, a basal forebrain structure involved in reinforcement learning[64]. Few brain regions received enriched projections from CeM[Sst] neurons. Among them, the lateral and medial geniculate complex (LG, MG), thalamic nodes involved in defensive behaviors, and the ventral posterolateral/medial nucleus of the thalamus (VPL/M), a relay for somatosensory signals[65–69].

The PBN is an interesting output region because it receives projections from CeM[Htr2a] neurons driving food consumption, from CeM[Sst] neurons driving water consumption, and from CeM[Dlk1] neurons suppressing food intake during nausea[70]. It is likely that CeM[Htr2a] → PBN and CeM[Sst] → PBN projectors target different microcircuits in the PBN, since their optogenetic activation elicited different behaviors. Anorexigenic CeM[Dlk1] → PBN projectors may interfere with the appetitive CeM→PBN projectors, for example, through a presynaptic inhibition mechanism. Future work is needed to work out the information flow from the CeM to the PBN.

In conclusion, the present manuscript provides a detailed functional characterization of molecularly- and anatomically-defined appetitive CeA subpopulations. Our findings indicate that neurons driving food or water consumption are confined to the CeM and that separate CeM subpopulations regulate water only (CeM[Sst]), versus water or food consumption (CeM[Htr2a]). The response properties of these CeM neurons show interesting differences regarding reward value and physical attributes of the stimuli. In the future, further characterization of these circuits, including modulation by neuropeptides and comparative evolutionary studies in healthy and diseased subjects may provide insights into the etiology of eating and drinking disorders and may help to develop therapeutic strategies to combat these common problems.

## Methods

### Animals
Experiments were always performed using adult mice (>12 weeks). Mice are group-housed under standard laboratory conditions and maintained under a 12-h light/12-h dark cycle with food and water ad libitum. The Htr2a-Cre BAC transgenic line (stock Tg(Htr2a-Cre) KM208Gsat/Mmucd) was imported from the Mutant Mouse Regional Resource Center 482 (https://www.mmrrc.org/). SOM-IRES-cre (SSTtm2.1(cre)Zjh/J) mice were acquired from the Jackson Laboratory. Ai9lsl−TdTomato (B6.Cg- Gt(ROSA) 26Sortm9(CAG- tdTomato) Hze/J)[71], FPDI (B6;129S6Gt(ROSA)26Sortm9 (CAG-mCherry, -CHRM4*) Dym)[35] mouse lines were as described previously. All mice were backcrossed into a C57BL/6NRj background (Janvier Labs−http://www.janvier-labs.com). Both male and female mice were used and all the experiments were performed following regulations from the government of Upper Bavaria.

### Generation of Wfs1-FlpoER transgenic mice
For the generation of Wfs1-FlpoER mice, a BAC homologous recombination method was used. The BAC clone RP23-405O19 (CHORI) was targeted with a FLPoERT2 expression cassette[72] and a WPRE sequence followed by the bovine growth hormone polyadenylation signal (pA) and a kanamycin resistance cassette. The BAC homologous recombination cassette was assembled in the pcDNA3.1 (+) vector (Invitrogen) as follows: Homology arms A and B were designed to flank mouse Wfs1 exon 2. A construct containing the homology arm A and the NLSFlpoERT2WPREpA sequences was synthetized by Eurofins and cloned into the pUC57 vector (GenScript). Homology arm B was synthetized by PCR using the following primer sets: 5′-CGA-TATCAACTCAGGCACC-3′ (forward primer for arm B), 5′-AATCTCGAGCAGGGACACTG-3′ (reverse primer for arm B). First, pCDNA3.1 (+) was digested with NheI/AflII to insert the homologous arm A − NLSFlpoERT2WPREpA fragment into this site. Second, the kanamycin resistance cassette (Gene Bridges GmbH) was cloned into the AflII/EcoRV site before the arm B. Then, the homologous arm B was inserted into the EcoRV/XhoI site. This targeting vector was digested with NheI/XhoI followed by purification of the insert on a 0.7% agarose gel. Homologous recombinant BACs were obtained using established methods[73,74], screened by PCR, and verified by Southern blotting. Modified BAC DNA was prepared using the large construct DNA

purification kit−NucleoBond® Xtra BAC (Macherey-Nagel), linearized with PmeI and purified through a Sepharose™ separation column[75]. BAC DNA (2.2 ng/µl) was injected into pronuclei of fertilized oocytes of C57BL/6 mice. BAC transgenic mice were identified by PCR using the primers 5′ GCTCTATTCAGGACATTTTCACATCTCTAC 3′ and 5′ CCTCTCGAATCTCTCCACGAAC 3′. The Wfs-FlpoER transgene was inserted in chromosome 15 between positions 11,537,026 and 11,537,028 (mm10 *Mus musculus* reference genome). A small 6 bp genomic duplication was found at the integration site. According to RefSeq, there are no genes annotated in this region.

### Generation of PKCδ-Flpo transgenic mice

A recombineering protocol was used to insert an NLSFLPo-pA expression cassette[76] into the ATG start site of the PKCdelta locus on the BAC clone RP23-283B12 (CHORI). The wild-type loxP site present in the RP23 BAC backbone was replaced by a piggyBAC-ampR cassette. Plasmid construction and BAC modification was done by Gene Bridges GmbH. BAC DNA for pronuclear injection was prepared using the large construct DNA purification kit−NucleoBond® Xtra BAC (Macherey-Nagel), linearized with PI-SceI and purified over a home-made Sepharose CL4B column (Johansson et al.[75]). Fractions were analyzed by pulsed-field gel electrophoresis to identify the sample with the highest concentration of linearized BAC DNA and lowest concentration of vector DNA. Linearized BAC DNA was injected at a concentration of 3.65 ng/µl into pronuclei of fertilized oocytes of C57BL/6 mice. BAC transgenic mice were identified by PCR using the primers 5′ AAACTGCATCACCTTCTCACATCTCC 3′ and 5′ CTCTCGAATCTCTC-CACGAACTGC 3′. The Pkcδ-Flpo transgene was inserted in chromosome 15 between positions 21,426,977 and 21,427,824 (mm10 *Mus musculus* reference genome), leading to an 846 bp genomic deletion of the host genome. According to RefSeq, intron 3 of *Cdh12* is annotated in the deleted region.

### Viral constructs

The following AAV viruses were purchased from the University of North Carolina Vector Core (https://www.med.unc.edu/genetherapy/vectorcore): AAV5-Ef1a-DIO-eNpHR3.0-mCherry, AAV5-Ef1a-DIO-mCherry, AAV5-Ef1a-DIO-hChR2(H134R)-EYFP-WPRE-pA, AAV5-Ef1a-DIO-EYFP-WPRE-pA, AAV5-hSyn-Con/Fon-hChR2(H134R)-EYFP-WPRE, AAV5-hSyn-Con/Fon-EYFP-WPRE, AAV5-hSyn-Con/Foff-hChR2(H134R)-EYFP-WPRE, AAV5-hSyn-Con/Foff-EYFP-WPRE, AAV5-EF1a-fDIO-hChR2 (H134R)-EYFP-WPRE.

AAV8-nEF-Con/Foff iC++-EYFP, AAV8-nEF-Con/Foff-EYFP and AAV8-EF1a-Con/Foff-GCaMP6m viruses were generated as described[32].

### Viral injections

Mice were anaesthetized using isoflurane (Cp-pharma) and placed on a heating pad on a stereotaxic frame (Model 1900−Kopf Instruments). Carprofen (Rimadyl−Zoetis) (20 mg/kg body weight) was given via subcutaneous injection. Mice were bilaterally (or unilaterally for calcium imaging experiments) injected using glass pipettes (#708707, BLAUBRAND intraMARK) with 0.3 µl of virus in the CeA by using the following coordinates calculated with respect to bregma: for the CeA and CeL: −1.20 mm anteroposterior, ±2.87 mm lateral, −4.65 to −4.72 mm ventral; for the CeM −1.155 mm anteroposterior, ±2.87 mm lateral, −4.65 to −4.72 mm ventral. The following stereotaxic coordinates were used for the PBN: −5.2 mm anteroposterior, ±1.4 mm lateral, −3.85 mm ventral. Virus was allowed to be expressed for a minimum duration of 3 weeks before histology or behavioral paradigms. For animals not undergoing implant surgery, the incision was sutured.

### Optic fiber implants

Mice used in optogenetic experiments were implanted with optic fibers (200-µm core, 0.22 NA, 1.25-mm ferrule−Thorlabs) above the CeA (−4.35 mm ventral from bregma) or the PBN (−3.6 mm ventral from bregma) immediately after viral injection. The skull was first protected with a layer of histo glue (Histoacryl, Braun), the fibers were then fixed to the skull using UV light-curable glue (Loctite AA3491−Henkel), and the exposed skull was covered with dental acrylic (Paladur−Heraeus).

### GRIN lens implantation and baseplate fixation

Three weeks after GCaMP6m viral injection in the CeA, mice were implanted with a gradient-index (GRIN) lens. At the same coordinates of the injection, a small craniotomy was made and a 20 G needle was slowly lowered into the brain to clear the path for the lens to a depth of −4.5 mm from bregma. After retraction of the needle, a GRIN lens (ProView lens; diameter, 0.5 mm; length, ~8.4 mm, Inscopix) was slowly implanted above the CeA and then fixed to the skull using UV light-curable glue (Loctite AA3491−Henkel). The skull was first protected with histo glue (Histoacryl, Braun), and the implant fixed with dental acrylic (Paladur−Heraeus). 4–8 weeks after GRIN lens implantation, mice were "baseplated" under anesthesia. Briefly, in the stereotaxic setup, a baseplate (BPL-2; Inscopix) was positioned above the GRIN lens, adjusting the distance and the focal plane until the neurons were visible. The baseplate was fixed using C&B Metabond (Parkell). A baseplate cap (BCP-2, Inscopix) was left in place to protect the lens.

### Tamoxifen

Tamoxifen was prepared by dissolving 20 mg tamoxifen in 100% ethanol to obtain a final concentration of 40 mg/ml. The solution was then diluted 1:1 with Kolliphor ® EL (Sigma). By heating and stirring the mixture, the ethanol evaporates. After that, 100 µl aliquots were frozen until use. Before use, the stock solution was diluted with PBS to the desired concentration and the pH value of 7.4 was confirmed. Mice were injected with ~200 µl of tamoxifen solution (200 mg per kilogram of body weight) for 3 days every other day. The mice that underwent brain viral injection received the first injection immediately after the surgery.

### Behavioral assays

Mice were bilaterally tethered to optic fiber patch cables (Doric Lenses or Thorlabs) via a mating sleeve (Thorlabs). The patch cables were connected via a rotary joint (Doric Lenses) to a 473 nm or 561 nm (CNI lasers) laser. Photoactivation and photoinhibition experiments were conducted with 10–15 mW 10 ms, 473 nm light pulses at 20 Hz, using a pulser (Prizmatix) controlled by the Ethovision software XT 14 (Noldus), or 561 nm 15 mW constant light.

**Drinking behavior.** For water deprivation experiments, mice were water deprived and the following day trained to drink in a 32 × 35 cm plastic arena for 30 min. This training was mainly done to habituate the mice to the new setup and train them to drink from specific pipettes. For photoactivation experiments, mice were then tested the following two days in the setup with access to two pipettes of water with a sequence of 10 min laser OFF/10 min laser ON (10 min laser ON/10 min laser OFF for PKCδ mice), or simply for 30 min laser ON. The two experiments were performed in a randomized order. For experiments in normal conditions, the same protocols were used, but the mice had *ad libitum* water in their cage. For photoinhibition experiments, water-deprived mice had access to water for 30 min while photoinhibited the day after the training. The amount of water was manually measured.

**Feeding behavior.** The experiments were conducted in a 32 × 35 cm plastic arena containing two plastic cups in opposite corners, one with dustless precision pre-weighed food pellets (20 mg each) (Bio-Serv-F0071) and one empty. Experiments were conducted over 40 min sessions and the remaining food was weighed. The day before the experiment, mice were habituated to the new food by placing some of these pellets into the cage together with the normal food.

**Palatable reward consumption.** Mice were food deprived for 16 h and allowed to consume a palatable reward solution (Fresubin, 2 kcal/ml) for 30–45 min. The mice went back to *ad libitum* food and the following day were tested for 30 min for Fresubin consumption while photoinhibited.

**Real-time place preference.** ChR2-expressing mice and corresponding controls were allowed to freely navigate in a custom-made plexiglas two-chambered arena ($50 \times 25 \times 25$ cm) for 20 min. ChR2-expressing mice and controls received 20 Hz 473 nm photostimulation in one compartment. The experiment was repeated for two consecutive days alternating the photostimulated and neutral chambers. Preference index = percentage of time spent in the photoactivated chamber.

**Conditioned flavor preference.** Mice were water deprived and for two consecutive days were given the choice between two nonnutritive flavored liquids (0.3 % grape or cherry and 0.15% saccharin) for 30 min. The preferred taste was defined as the taste from the two that the mice drank more of. Conditioning was conducted over four consecutive days with two sessions per day. The least preferred taste was paired with optogenetic activation. In conditioning session one, the least preferred taste was paired with light for 15 min. In conditioning session two, the mice were presented with the more preferred taste in the absence of photostimulation. The order of the sessions was inverted each day, occurring 4–6 h apart. Conditioned flavor preference was tested the day after the final conditioning session, when the mice were presented with both tastes. Preference index = (Preferred taste)/(Preferred + least preferred taste).

**Quinine consumption.** Mice were water deprived and the following day trained to drink in a $32 \times 35$ cm plastic arena for 30 min. The day after, mice were tested in the setup with access to two pipettes containing a 10 mM or 100 µM quinine solution for 30 min. The amount of quinine was manually measured.

**Open field.** Mice were allowed to explore a custom-built plexiglas arena ($40 \times 40 \times 25$ cm) for 10 min. During the whole experiment, mice received photoactivation or photoinhibition.

## In vivo calcium imaging of freely moving mice

All in vivo imaging experiments were conducted on freely moving mice. GCaMP6m fluorescence signals were acquired using a miniature integrated fluorescence microscope system (nVoke–Inscopix) secured in the baseplate holder before each imaging session. Mice were habituated to the miniscope procedure for 3 days before behavioral experiments for 30 min per day. Settings were kept constant within subjects and across imaging sessions. Image acquisition and behavior were synchronized using the data acquisition box of the nVoke Imaging System (Inscopix) triggered by the Ethovision XT 14 software (Noldus) through a TTL box (Noldus) connected to the USB-IO box from the Ethovision system (Noldus). For drinking experiments, mice were water deprived and trained the next day to drink in a $32 \times 35$ cm plastic arena for 30 min. Calcium activity was recorded the following day during a 10-min period of habituation without water and a subsequent 10-min period of exposure to water. For feeding experiments, mice were food restricted and trained the following day to consume dustless precision pre-weighed food pellets (20 mg each) (Bio-Serv- F0071) in a $32 \times 35$ cm plastic arena containing two plastic cups in opposite corners, one of which empty. The next day, after 10 min of recording without food, animals were exposed to food for an additional 10 min and their calcium activity was recorded. The Fresubin experiment was conducted as previously described in this manuscript. On the test day, the calcium activity of fed mice was recorded during a 10-min period of habituation

followed by a 10-min period after the introduction of Fresubin. To study water/saccharin/quinine behavior, mice were water deprived and habituated the following day to water administration in the experimental setup. On the next day, the experiment was carried out in three consecutive 10-min sessions. During the first, second, and third session, water, saccharin (0.15%), and quinine (10 mM) were respectively administered at minutes 1 and 6. For imaging data processing and analysis we used IDPS (Inscopix data 721 processing software) version 1.8.0.

For the stable-unstable and generalizer-specializer analysis we calculated the percentages of neurons per animal that were maintaining a positive or negative correlation (pos–pos, neg–neg: stable or generalizer) or that were changing the correlation (neg–pos, pos–neg, neg–neutral, pos–neutral, neutral–neg, neutral–pos: unstable or specializer) between the two stimuli.

## CEBRA analysis

To understand if the recorded behaviors contributed to changes in neural activity, we applied the CEBRA pipeline for feature selection. If certain behaviors account significantly for neural activity, we would have expected to see clear structures in the CEBRA embeddings compared to shuffled neural data. In brief, we chose the CEBRA Hybrid model (self-supervised) with parameters used in this study as below:

Hybrid model, model_architecture = "offset5-model-mse", conditional = "time-delta", hybrid = True, distance = "euclidean", batch_size = 600–1200, learning_rate = 1e-4. Iteration = 8000.

To examine the association between CEBRA embeddings and behaviors we applied the Random Forest (RF) classifier for behavioral decoding. In brief, we fitted behavioral labels with our CEBRA embeddings, we used 70% of the data as training data and 30% of the data as test data (with data splitting in random orders). In addition, we permuted the data 1000 times randomly and tested it with the trained RF classifier. We inspected the model by checking the decoding accuracy and the out of bag (OOB) error. To prevent overfitting of the model, we decoded behavioral labels first within the same animal (Figs. 6f, g, i, j and 7e; Supplementary Figs. 6d, f and 7e), then we decoded the behavioral labels across animals (Figs. 6f, g and 7e; Supplementary Figs. 6d and 7e).

## Brain-slice preparation and electrophysiological recordings

Mice were anesthetized using isoflurane and subsequently sacrificed by decapitation. The brains were then placed in a cutting solution composed of 95% oxygen and 5% carbon dioxide, with a mixture of 30 mM NaCl, 4.5 mM KCl, 1 mM $MgCl_2$, 26 mM $NaHCO_3$, 1.2 mM $NaH_2PO_4$, 10 mM glucose, and 194 mM sucrose. The brains were sliced to a thickness of 280 µm using a Leica VT1000S vibratome (Germany) and transferred to an artificial cerebrospinal fluid (aCSF) solution containing 124 mM NaCl, 4.5 mM KCl, 1 mM $MgCl_2$, 26 mM $NaHCO_3$, 1.2 mM $NaH_2PO_4$, 10 mM glucose, and 2 mM $CaCl_2$ (310–320 mOsm), saturated with 95% $O_2$/5% $CO_2$ at 32 °C for 1 h before being brought to room temperature. The brain slices were then placed in a recording chamber that was continuously perfused with aCSF solution, also saturated with 95% $O_2$/5% $CO_2$, at a temperature of 30–32 °C.

Whole-cell patch-clamp recordings were carried out as previously described[21]. Patch pipettes were fabricated from filament-containing borosilicate micropipettes (World Precision Instruments) using a P-1000 micropipette puller (Sutter Instruments, Novato, CA) to achieve a resistance of 6–8 MΩ. The intracellular solution consisted of 130 mM potassium gluconate, 10 mM KCl, 2 mM $MgCl_2$, 10 mM HEPES, 2 mM Na-ATP, 0.2 mM $Na_2GTP$ at a pH of 7.35, and an osmotic pressure of 290 mOsm. The brain slices were visualized using a fluorescence microscope equipped with IR-DIC optics (Olympus BX51). The holding potential for excitatory postsynaptic currents (EPSC) was set at −70 mV and 0 mV for inhibitory postsynaptic currents (IPSC). Data was

acquired using a MultiClamp 700B amplifier, Digidata 1550 digitizer (Molecular Devices), and analyzed using Clampex 10.3 software (Molecular Devices, Sunnyvale, CA). The data was sampled at 10 kHz and filtered at 2 kHz, and further analyzed using Clampfit (Molecular Devices). A similar strategy to evaluate the connections inside the CeA was recently used for CeA-Dlk1 neurons[70].

For optogenetic studies, neurons were stimulated through the use of a multi-LED array system (CoolLED) connected to an Olympus BX51 microscope.

## Histology
Mice were anesthetized with ketamine/xylazine solution (Medistar and Serumwerk) (100 mg/kg and 16 mg/kg, respectively) and transcardially perfused with phosphate-buffered saline (PBS), followed by 4% paraformaldehyde (PFA) (1004005, Merck) (w/v) in PBS. Extracted brains were post-fixed overnight at 4 °C in 4% PFA (w/v) in PBS. Brains were either embedded in 6% agarose and sliced using a Vibratome (VT1000S −Leica) or incubated overnight in 25% sucrose, snap frozen in dry ice and sliced using a cryostat (Leica) into 50-µm free-floating coronal sections.

## Immunohistochemistry
Brain sections were blocked at room temperature for 2 h in 5% donkey serum (Biozol JIM-017-000-121) diluted in 1X PBS 0.5% TritonX-100 and then incubated with primary antibody at 4 °C overnight in the same solution. Primary antibodies: goat anti-mCherry (1:1000) (Origene AB0040-200), chicken anti-GFP (1:500) (Aves, GFP-1020), rabbit anti-Wfs1 (1:200) (made by the lab of Prof. Dr. Jens F. Rehfeld, University of Copenhagen, Denmark), mouse anti-PKCδ (1:100; 610398, BD Biosciences). After incubation with primary antibody, the sections were washed in 1X PBS (3 × 10 min) and incubated in secondary antibody in 1X PBS 0.5% TritonX-100 at 4 °C overnight. Secondary antibodies: Alexa Fluor donkey anti-rabbit/goat/chicken 488/Cy3/647, (1:300) (Jackson). After 3 × 10 min washes in 1X PBS, sections were incubated in DAPI and coverslipped (Dako).

## Microscopy and image processing
A Leica SP8 confocal microscope equipped with a 20×/0.75 IMM or 10×/0.30 FLUOTAR objective (Leica) was used to acquire Fluorescence z-stack images. For projection analysis, images were acquired at the same fixed exposure time and gain using a Leica Thunder microscope and the tile scan and automated mosaic merge functions of Leica LAS X software. Images were minimally processed with ImageJ software (NIH) to adjust for brightness and contrast for optimal representation of the data. For all cell quantifications of brain sections, ImageJ was used to manually count the cells. For the quantification of axon projections we calculated the corrected total fluorescence (CTF) adapted from this protocol (CTCF): https://theolb.readthedocs.io/en/latest/imaging/measuring-cell-fluorescence-using-imagej.html. In brief, on each slide that was used for quantification, we measured the intensity of a square in the ROI (Region of Interest) and the background intensity. The CTF was quantified as: Integrated Density−(Area of selected ROI × Mean fluorescence of background readings). Afterward, the CTFs of the same ROI were averaged and normalized to the injection site. The normalization was performed as: Normalized intensity= IntROI/IntInjectionSite.

## Statistical analysis
No statistical methods were used to pre-determine sample sizes. The numbers of samples in each group were based on those in previously published studies. Statistical analyses were performed with Prism 9 (GraphPad) and all statistics are indicated in the figure legends. T-tests or two-way ANOVA with Bonferroni post-hoc tests were used for individual comparisons of normally distributed data. Normality was assessed using Shapiro-Wilk test. If the data were not normally distributed, Mann-Whitney U test and Wilcoxon signed-rank test were performed for individual comparisons. After the conclusion of experiments, virus expression and implants placement were verified. Mice with very low or null virus expression were excluded from analysis.

## Reporting summary
Further information on research design is available in the Nature Portfolio Reporting Summary linked to this article.

## Data availability
Source Data are provided with this paper. Additional raw data from this study are available from the corresponding author upon request. They are not deposited to a public database due to their large size and size limitations of online depositories.

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

## Acknowledgements

We thank Soo Jin Min-Weissenhorn and the Transgenic Service for help with generating transgenic mouse lines; Jens F. Rehfeld, University of Copenhagen, for providing the Wfs1 antibody; Minh Chau Mai, Emily Russell, Amelie Neubauer for help with management of the animal colonies; Alessandra Monaco for support in cryosectioning; and the company Cergentis B.V. for chromosomal mapping of the Flpo transgenic lines. This study was supported by the Max-Planck Society and the European Research Council under the European Union's Horizon 2020 research and innovation program (No. 885192; to R.K.).

## Author contributions

F.F. and R.K. conceptualized the study and designed experiments. F.F. performed most of the experiments. S.C. performed CEBRA and projections intensity analyses. Y.M. assisted with stereotaxic surgeries and the quinine consumption experiments. C.P. performed and analyzed electrophysiology experiments. L.G. assisted with histology, immunohistochemistry, microscopy, and image processing. P.L.A.M. generated the Wfs1-FlpoER transgene, helped designing the PKCδ-Flpo transgene and establishing both transgenic mouse lines. C.R. and K.D. provided intersectional viruses. F.F. and R.K. wrote the manuscript with input from all authors. R.K. supervised and provided funding.

## Funding

## Competing interests

The authors declare no competing interests.
