## [Transparent Peer Review file · Nature Communications]

Food and water intake are regulated by distinct central amygdala circuits revealed using intersectional genetics

Corresponding Author: Professor Rüdiger Klein

Version 0:

Reviewer comments:

Reviewer #1

(Remarks to the Author)

In this study, Fermani et al. used intersectional genetics to independently manipulate four neuronal subpopulations in lateral (CeL) and medial (CeM) aspects of the mouse central amygdalar nucleus. Of interest, the authors reported that neurons involved in water and/or food intake are confined to the CeM. More specifically, separate CeM subpopulations mediated water consumption (somatostatin, CeM^{Sst}), and water or food consumption (serotonin-sensitive, CeMHtr2a). CeM subpopulations also induced real-time place preference. Consistently, in vivo calcium imaging studies revealed that CeM^{Htr2a} and CeM^{Sst} neurons respond to reward consumption. The authors concluded that CeM microcircuits can detect and evaluate liquid and solid reinforcing stimuli.

This is an interesting study that clarifies the behavioral functions of diverse cell types in the central amygdala, particularly the poorly understood involvement of central amygdala neurons in appetitive behaviors. The application of novel intersectional strategies is fairly elegant, and overall, the study constitutes a relevant and valuable contribution to the field. However, I have two specific observations.

In Methods, 612-614, the authors explain that they used two sets of stereotaxic coordinates, one for the entire CeA and CeL, and the other for the CeM. How confident are the authors that their activation or imaging studies covered the entire CeA when they intended to do so by aiming at CeL? Was activation of CeL restricted to CeL proper? A description of the rationale behind this approach will improve the paper. This is relevant when considering that the intersectional approach does not separate completely CeM from CeL neurons.

The authors distinguished between populations, CeM^{Htr2a} and CeM^{Sst}, that respond to water only versus to water and food. Did the authors observe differential changes in e.g. tongue movement when turning on these populations?

Reviewer #2

(Remarks to the Author)

In this manuscript from Fermani et al. the authors examine an important question regarding the valence and reward processing within genetically defined populations (Htr2a and Sst) in the CeA. Interestingly, they find that while there are fewer neurons in of these populations in the medial CeA, these are the neurons that are conveying positive valence properties both when stimulated and when recorded from using calcium indicators. This work is very important and delves into nuance with regard to both intra CeA circuits and projection targets. There remain some important considerations and clarifications that the authors should consider.

Statistics Overall: there needs to be some better description for when Mann-Whitney U tests and Wilcoxon signed-rank tests were used vs. t-tests. It is stated that this is when "normality was not assumed" but were these data sets determined to not be normal?

For figure 1 it would be beneficial to see representation across the CeA. The authors do a good job highlighting more anterior and posterior parts, but it would be nice to see how these neurons are represented across the CeA as it is quite a long structure and the lateral and medial parts in particular shift along these structures. The way the data is represented in Fig 1 H, L, P T, is a bit confusing and hard to interpret.

Retrograde labeling has demonstrated that the majority of the cells projecting to the PBN, however, derive from the lateral CeA...it would be nice to see retrograde labeling in the Cre-on/Flp-off intersectional strategies here. This is especially interesting given the later figures when the PBN is examined more intensely, so I would suggest putting that later.

There are definitely green Chr2 positive fibers in the lateral CeA of the Cre-on/Flp-off animals ...do these neurons form projections within the CeAL and form connections?

Please further define how the water deprivation was performed and how long the animals were without water and if they had other time to drink after the experiment before water was again removed (figure 2 expts) – that's a very large amount of water to drink. Even if it was being driven, were there any aversive effects / later differences in groups? Consuming a days' worth of water (being driven especially) in that short seems like it could have downstream effects

- How was drinking verified – was there a drip cage or any non-animal control? Lickometers? How did they measure water intake every 10 min in grams?
- Were the same animals that drank more in water deprived/ad lib conditions?
- Were the same animals drinking more after each opto stim period, or was there any rebound from drinking a lot and then having perceived that consumption a second later after stim ceased?

While both values are significant for Htr2a in Fig 3D-E, I am wondering if it would be beneficial to calculate Cohen's D here because it seems the effect size is bigger in the IC++ experiment. It is probably problematic to directly compare because of the 2 different opsins, but interesting nonetheless? Referring to IC++ as a "blue-shifted halorhodopsin" is not entirely accurate as one is an ion pump and one is a channel (though the activation wavelength difference is correct).

For the preference index in the RTPP experiments (Fig 4) please draw a dashed line at 50% so that the reader can have that context when looking at the data. Also it would be good to keep preference index either a % or a scale of 0-1 in the same figure.

With regard to Fig 5 using the phrase "stimulating" (especially right after doing RTPP opto in Fig 4). Is somewhat confusing to the reader, perhaps there is a better way to phrase this?

With regard to the specializer vs. generalizer cells in Fig 6, it is slightly problematic that the authors did not use a more aversive dose of quinine. What tests were conducted to determine if 10 uM quinine was aversive? In previous studies following surgical manipulation it has been noted that there is a shift in the quinine DR curve, where concentrations 10 uM and lower did not result in aversion behavior (see Torruella-Suarez et al., 2020.) Was the tastant ordering counterbalanced for presentation?

Figure 8 is a tour de force and nicely brings together the main concepts of the paper. Still it would be interesting to see what would happen with the CeAL projection neurons to the PBN, especially given the differences in terminal expression within the PBN! I feel this experiment, maybe even if just completed for a limited set of endpoints would be valuable to the field. I would put the Sst images in the main figure and not bury them in the supplement. Especially because the representative image for the PBN looks somewhat strange from the Sst CeM?

Some of the interpretations maybe overstated and could be dialed back a bit. For example, the authors are trying to show that CeL neurons are not rewarding in RTPP but are in conditioned flavor preference – I think there might be a less strong effect (there appears to be trends in Fig 4C and single animals in the control and Chr2 conditions for Htr2a driving effects), but still a possible effect. They even say in the discussion that "the inferior performance of the Con/Foff virus may have resulted from incomplete coverage of Flp expression in CeL neurons." Could this not be part of the explanation for why the trend looks the same, but isn't as important as CeM. While it may be equally/more important for drinking than CeM, the authors may be over interpreting the differences here.

Reviewer #3

(Remarks to the Author)

Summary

One of the hallmarks of CeA biology is the extensive heterogeneity of CeA neurones at multiple anatomical, gene expression, and physiological levels. The development of a FlpO driver to mark CeL neurones is a major and important advance to allow for intersectional targeting to characterize heterogeneity. The authors perform a number of important experiments to dissect essentially four mostly distinct and enriched populations of CeA neurones. The identification of new behaviors and encoding (particularly Fig. 5/6) is an important step forward for the field and this lab going back to the original findings in Dougless et al, NN. 2017. Overall, I like this paper and have made some suggestions below that I hope will improve things. In some cases these are dealbreakers. In others I spell out what are not dealbreakers, but in hopes that this will improve future experiments and ongoing projects.

Major revision points

Overall: The authors have done a heroic amount of work and made some important advances. I encourage them, in the future, to endeavour to run more powered studies where exploration of sex differences is possible and to plot their data in a

way that allows the reader to appreciate potential sex differences. I recognize it would be a monumental amount of work to redo all of their figures in this way so it is not a requirement for my potential and ultimate acceptance of this manuscript. Nonetheless, I highly encourage the authors to do this for future studies...even if it means the experiments take longer. In the figure legends, please list the number of animals for each group. If it is the same for each panel, it can be done at the end of the legend. In some places it was hard to gauge the rigor of the experiments due to the omission of this information.

Figure 1: The intersectional targeting approach is effective at producing an enrichment of CeL vs CeM Htr2a and Sst neurones. It is not perfect. This needs to be discussed as a potential limitation in the Discussion section. Also it is possible that the Wfs1-FlpOER line is not penetrant in all Wfs1+ cells (Extended Data 1k-o) contributing to the CeM intersectional targeting being slightly less specific. This is mentioned at line 420 somewhat. This does not undermine the study. Any other reviewer that suggests this is missing the forest for the trees.

Figure 2. An attempt to describe or depict the optical fiber placement of each mouse plotted should be shown here or in the extended data as well as an explanation of any animals removed from the data shown due to predetermined criteria. The data in this figure are very nice and impactful, but the rigor should be mentioned at some point. Also, no attempt was made to disentangle fluid vs calorificity vs palatability. This is not a dealbreaker, but promotion of fluid intake even in the face of subtle quinine adulteration would add that activation of these cells is stimulating a licking and fluid intake motor program or that its activation is capable of encoding valence for fluids. The conditioned taste experiment is consistent with this idea, but the lack of a negative flavor dissection in Fig. 2 is highlighted when it is included in Fig. 6. The data here suggests that Htr2a neurons may encode the valence of the fluid. Disentangling these possibilities with free consumption of quinine adulterated fluid with opto would be impactful.

Extended Data 4: Please show the distance traveled in the open field. Center time is difficult to interpret without it. If there is substantial variance, then the data can be shown as % of time spent in the center in which case Htr2a overall activation might trend towards anxiolysis but we can't determine this without distance traveled. Even if one of these cohorts did show anxiolysis recalculated as % time I don't think this undermines the study and is consistent with the RTPP. In other words, there is no reason not to show this.

Figure 5 K,I and 6 F,G: Is there a way to highlight what neurons are specializers in the chord diagrams or in the side bar graphs? Also do the individual data points represent the % of neurons recorded from an individual animal? The reader is having to infer a lot from these plots and the text. It's not overly clear how these experiments were done, so more detail can be added to the methods as well. As with the opto data, the histology of GRIN lens implanted animals or rigor thereof should be shown or detailed somewhere.

Figure 7: The electrophysiological studies could be much improved in this figure with regards to how they are explained and executed. There is no assessment of spike fidelity or latency in PKCdelta neurones and the authors say "providing evidence for a monosynaptic connection". There was no mention of drugs used to examine the specificity of optically induced IPSCs in the CeM (TTX, 4-AP) or blockade of the putative IPSC (GABAzine, picrotoxin). I agree that this all likely to be the case and perhaps unsurprising, but the rigor is lacking and data overstated. Please improve this.

Figure 8: An average intensity calculation of the output terminals at each region of CeM vs CeL Htr2a and Sst neurons would provide quantification and support for areas in which the CeM "covered a larger area within the target region" (line 357). I'd also like to see each of the areas shown if possible. The PVT enrichment for CeM Htr2a neurones is particularly intriguing but I don't see it in Figure 8 or Extended Data 7. As before, rigor of optical fiber placement should be shown or described (at least).

Minor revision points

Title: Why "uptake" instead of "intake"? The former is weird. Animals don't "take up" food or water, they "take in" food or water. This is used throughout the manuscript and it is distracting.

Introduction:

Line 47: CeA neurones can also be segregated and classified based on their physiological properties. See work from Yamar Carrasquillo.

Line 66-67: "fully worked out". Avoid colloquial phrases.

Line 82: "Unexpectedly". I understand using narrative adverbs to tell the story, but this is not unexpected in my opinion. Ip et al. Cell Metabolism 2019 (<https://pubmed.ncbi.nlm.nih.gov/31031093/>) was not cited. This is an important paper in adding to the idea that CeA neurones can actually promote feeding. Moreover, the localization of CeA NPY neurones to the CeM subdivision is very important to the overall conclusions of this study. It should be cited and discussed. This does not undercut the substantive impact and rigor of the current study to dissect CeL vs CeM subpopulations.

Results:

Line 170: Food "uptake".

Line 232: neuronal "activities". Neuronal "activity" is better. Multiple instances of this throughout the results section.

Line 238: "relation"

Line 366: This whole paragraph should be moved to the Discussion section.

Discussion: Fine

Methods: Fine, but see points above.

Overall decision: Please submit a revised manuscript for a second review addressing some of these comments.

Version 1:

Reviewer comments:

Reviewer #1

(Remarks to the Author)

I believe the authors have appropriately addressed the Reviewer's questions, providing a significantly improved and important manuscript.

Reviewer #2

(Remarks to the Author)

the authors have addressed the majority of my comments. I find this work to be compelling and well conducted, and support its publication at Nature Comms.

Reviewer #3

(Remarks to the Author)

One of the hallmarks of CeA biology is the extensive heterogeneity of CeA neurones at multiple anatomical, gene expression, and physiological levels. The development of a FlpO driver to mark CeL neurones is a major and important advance to allow for intersectional targeting to characterize heterogeneity. The authors perform a number of important experiments to dissect essentially four mostly distinct and enriched populations of CeA neurones. The identification of new behaviors and encoding (particularly Fig. 5/6) is an important step forward for the field and this lab going back to the original findings in Dougless et al, NN. 2017. In their resubmission, I feel that the authors were responsive and the manuscript still very much represents an important advance for understanding of the CeA. I think this revised version is appropriate for publication at Nature Communications.

Response to reviewers' comments

Reviewer #1 (Remarks to the Author):

In this study, Fermani et al. used intersectional genetics to independently manipulate four neuronal subpopulations in lateral (CeL) and medial (CeM) aspects of the mouse central amygdalar nucleus. Of interest, the authors reported that neurons involved in water and/or food intake are confined to the CeM. More specifically, separate CeM subpopulations mediated water consumption (somatostatin, CeM^{Sst}), and water or food consumption (serotonin-sensitive, CeMHtr2a). CeM subpopulations also induced real-time place preference. Consistently, in vivo calcium imaging studies revealed that CeM^{Htr2a} and CeM^{Sst} neurons respond to reward consumption. The authors concluded that CeM microcircuits can detect and evaluate liquid and solid reinforcing stimuli.

This is an interesting study that clarifies the behavioral functions of diverse cell types in the central amygdala, particularly the poorly understood involvement of central amygdala neurons in appetitive behaviors. The application of novel intersectional strategies is fairly elegant, and overall, the study constitutes a relevant and valuable contribution to the field. However, I have two specific observations.

In Methods, 612-614, the authors explain that they used two sets of stereotaxic coordinates, one for the entire CeA and CeL, and the other for the CeM. How confident are the authors that their activation or imaging studies covered the entire CeA when they intended to do so by aiming at CeL? Was activation of CeL restricted to CeL proper? A description of the rationale behind this approach will improve the paper. This is relevant when considering that the intersectional approach does not separate completely CeM from CeL neurons.

We thank the reviewer for their positive comments on our study. Regarding stereotaxic injections, the difference in anterior-posterior coordinates between the CeL and CeM is minimal (-1.155 for CeM and -1.22 for CeL). This variation reflects anatomical differences: the CeM is larger in the anterior region and smaller in the posterior, whereas the CeL shows the opposite pattern, being smaller in the anterior and larger in the posterior region. Using either of these coordinates allows us to effectively target the entire amygdala structure. However, we chose a slightly more anterior coordinate for CeM and posterior for CeL to follow the CeA anatomy. In Figure 1 below, we show examples of the entire CeA region from anterior to posterior in an Htr2a-Cre mouse injected with AAV-DIO-EYFP using the CeL coordinates. The pictures show complete labeling throughout the CeA, including the more anterior CeM region.

In the revised manuscript, we have added additional sections to revised main Figure 1, depicting the expression of intersectional viruses in the CeL and CeM of Htr2a-Cre::Wfs1-FlpoER and Sst-Cre::Wfs1-FlpoER mice. As indicated in the main figure, viral protein expression is displayed at Bregma -0.9 and -1.2 of the CeA, giving a better representation of expression across the CeA (main Figure 1e-r).

Fig.1 Representative images of the CeA from anterior (top left) to posterior (bottom right) of an Htr2a-Cre mouse injected with AAV-DIO-EYFP. Scale bar: 500 μm .

The authors distinguished between populations, $\text{CeM}^{\text{Htr2a}}$ and CeM^{Sst} , that respond to water only versus to water and food. Did the authors observe differential changes in e.g. tongue movement when turning on these populations?

We thank the reviewer for the interesting comment. Unfortunately, due to technical limitations, (not using a frontal camera and lickometer) we are unable to provide data that would address this comment. However, as suggested by reviewer #3, we have generated new experimental data to disentangle fluid intake versus valence detection. We included both $\text{CeM}^{\text{Htr2a}}$ and CeM^{Sst} subpopulations. Neurons were photoactivated during 30 min in which water-deprived mice were exposed to a 10 mM quinine solution. While photoactivated control mice avoided the bitter solution, activation of $\text{CeM}^{\text{Htr2a}}$ and CeM^{Sst} neurons stimulated quinine consumption. During subsequent days, we tested the same mice with a subtler quinine solution (100 μM) under water-deprived conditions for 30 minutes and observed that photoactivated ChR2-expressing mice consumed more fluid compared to controls. Comparing the intake of the two quinine concentrations by the same mice, we found that controls consumed similar small amounts of both solutions, whereas ChR2-expressing mice drank significantly more of the lower-concentrated solution (new Figure 6i,j).

These findings suggest that photoactivation of CeM^{Htr2a} and CeM^{Sst} neurons stimulates a fluid intake motor program, but that the mice are still able to distinguish varying concentrations of unpleasant taste. Considering that the neurons detect positive and negative valence for fluids, as suggested by the calcium data, and that neuron activation is rewarding as indicated by RTPP and self-stimulation (Douglass et al., 2017), we conclude that photoactivation of the neurons has two effects that contribute to fluid consumption: First, the activation of a fluid intake motor program that drives drinking independent of internal state and despite strongly aversive taste, and second, the generation of a rewarding effect that counterbalances the bitter taste. The stronger the bitter taste, the more it cancels out the rewarding effects and the less is consumed by the mice. We have added this discussion to the main text (line 502-510).

Reviewer #2 (Remarks to the Author):

In this manuscript from Fermani et al. the authors examine an important question regarding the valence and reward processing within genetically defined populations (Htr2a and Sst) in the CeA. Interestingly, they find that while there are fewer neurons in of these populations in the medial CeA, these are the neurons that are conveying positive valence properties both when stimulated and when recorded from using calcium indicators. This work is very important and delves into nuance with regard to both intra CeA circuits and projection targets. There remain some important considerations and clarifications that the authors should consider.

Statistics Overall: there needs to be some better description for when Mann-Whitney U tests and Wilcoxon signed-rank tests were used vs. t-tests. It is stated that this is when “normality was not assumed” but were these data sets determined to not be normal?

We thank the reviewer for requesting clarification on this point. We assessed normality with the Shapiro–Wilk test. If the data were normally distributed, we used t-tests, if not, we used Mann-Whitney U tests and Wilcoxon signed-rank tests. This information has now been clarified in Material and Methods (Statistical analysis).

For figure 1 it would be beneficial to see representation across the CeA. The authors do a good job highlighting more anterior and posterior parts, but it would be nice to see how these neurons are represented across the CeA as it is quite a long structure and the lateral and medial parts in particular shift along these structures. The way the data is represented in Fig 1 H, L, P T, is a bit confusing and hard to interpret.

We thank the reviewer for the suggestions. We added new panels to main Figure 1 showing representative sections from Bregma -0.9 and -1.2 of CeA depicting intersectional expression of EYFP across CeL and CeM subdivisions. We hope that this will be helpful.

We agree that the description of the quantifications of Figure 1 were a bit too short and unclear. We changed the wording of the Y axis to “Distribution of EYFP+ cells in CeL and CeM”. The revised results text was expanded explaining the data of every graph separately. This will hopefully be clearer.

Retrograde labeling has demonstrated that the majority of the cells projecting to the PBN, however, derive from the lateral CeA...it would be nice to see retrograde labeling in the Cre-on/Flp-off intersectional strategies here. This is especially interesting given the later figures when the PBN is examined more intensely, so I would suggest putting that later.

We thank the reviewer for the useful suggestion. We agree that retrograde labeling in the Cre-on/Flp-off intersectional strategies would provide valuable insights, especially given the focus on PBN projections in later figures. While we have not conducted the exact experiment proposed, we improved our tracing studies by quantifying projections from CeL/CeM Htr2a and Sst neurons to all major output targets, including the PBN. The results are presented in revised Fig. 8 and Extended data Figures 8 and 9. We described the results as follows:

“To determine the relative strength of the projections, we calculated the integrated fluorescence intensities in the output region normalized to background and injection sites. Our analysis revealed that for Htr2a neurons, the CeM fraction projected to a larger number of outputs compared to the CeL fraction (19 versus 8), whereas for Sst neurons, CeM and CeL fractions projected to similar numbers of outputs (20 versus 16) (Extended data Figs. 8, 9). A number of brain regions received strong projections from both Htr2a and Sst CeL and CeM fractions, including the bed nucleus of the stria terminalis (BNST), the lateral vestibular nucleus (LAV), the lateral and medial parabrachial nucleus (LPBN, MPBN), and the midbrain reticular nucleus (MRN) (Fig. 8a-d). The interstitial nucleus of the posterior limb of the anterior commissure (IPAC) and the substantia innominata (SI) received stronger projections from the CeM compared to the CeL fractions. Few brain regions received enriched projections from CeM^{Sst} neurons, including the lateral and medial geniculate complex (LG, MG) and the ventral posterolateral/medial nucleus of the thalamus (VPL/M) (Fig. 8 a-d; Extended data Fig. 8, 9). Several of these regions are known to be involved in rewarding and consummatory behaviors.”

There are definitely green ChR2 positive fibers in the lateral CeA of the Cre-on/Flp-off animals ...do these neurons form projections within the CeAL and form connections?

We do not have evidence that CeM neurons project to the CeL. The presence of green ChR2 positive fibers in the lateral CeA of the Cre-on/Flp-off animals is evidence for the system not working perfectly. Cells in the CeL that co-express Cre and Flpo will not express ChR2. However, CeL cells that only express Cre will express ChR2. Our quantifications revealed that the Cre-on/Flp-off virus results in an enrichment of ChR2-expressing cells in the CeM. Depending on the Cre driver, we target between 2.3- and 2.7-times more CeM than CeL cells. In comparison, the Cre-on/Flp-on virus works better. We see an enrichment of 4- and 10-times in CeL versus CeM. The strength of the system relies on the combination of the two viruses with photoactivation and -inhibition experiments.

Please further define how the water deprivation was performed and how long the animals were without water and if they had other time to drink after the experiment before water was again removed (figure 2 expts) – that’s a very large amount of water to drink. Even if it was being driven, were there any averse effects / later differences in groups? Consuming a days’ worth of water (being driven especially) in that short seems like it could have downstream effects

· How was drinking verified – was there a drip cage or any non-animal control? Lickometers? How did they measure water intake every 10 min in grams?

Mice were water-deprived in their home cages overnight (16-20 hours). On the following day, they were moved to a different arena and trained to drink water for 30 minutes. This training was mainly done to habituate the mice to the new setup and train them to drink from specific pipettes. For photoactivation experiments, mice were tested the following two days in the setup with access to water with a sequence of 10 min laser OFF/10 min laser ON, or simply for 30 min laser ON. For PKC δ mice, the sequence was reversed: 10 min laser ON/10 min laser OFF. The two experiments were performed in a randomized order. After the last experiment, mice had access to water ad libitum. For the photoinhibition experiment, the mice were tested the day after the training for 30 min laser ON. After the test, mice had access to water ad libitum. The water deprivation procedure is described in the Materials and Methods section.

We did not observe any adverse effects following photostimulation or drinking. To measure water intake, we used custom-made pipettes, constructed using spouts from the bottles typically used in the home cages, attached to 10 ml pipettes (see figure 2 below). At the start of each experiment, the pipettes were weighed on a scale, and at the end of each 10-minute or 30-minute session, we re-weighed the pipettes. The difference in weight provided the amount of water consumed.

Fig.2 Image of the pipette used during the drinking behavior experiments.

· Were the same animals that drank more in water deprived/ad lib conditions?

We examined the correlation between the amount of water consumed by the same mice under water-deprived (WD) and normal conditions (NC) (Figure 3 below). We did not find significant correlations.

Fig.3 Correlation analysis of water intake under different conditions.

a. Correlation between the amount of water consumed by the same Htr2a-Cre mice expressing ChR2 in CeA^{Htr2a} during 30 min photostimulation in water deprived (WD) and normal conditions (NC) (Correlation $r = 0.4102$, $p = 0.2729$). ($n = 9$ mice).

b. Correlation between the amount of water consumed by the same Sst-Cre mice expressing ChR2 in CeA^{Sst} during 30 min photostimulation in water deprived and normal conditions (Correlation $r = -0.2285$, $p = 0.4992$). ($n = 11$ mice).

c. Correlation between the amount of water consumed by the same Htr2a-Cre;Wfs1-FlpoER mice expressing ChR2 in CeM^{Htr2a} during 30 min photostimulation in water deprived and normal conditions (Correlation $r = -0.1210$, $p = 0.7565$). (n = 9 mice).

d. Correlation between the amount of water consumed by the same Sst-Cre;Wfs1-FlpoER mice expressing ChR2 in CeM^{Sst} during 30 min photostimulation in water deprived and normal conditions (Correlation $r = -0.3901$, $p = 0.2994$). (n = 9 mice).

Were the same animals drinking more after each opto stim period, or was there any rebound from drinking a lot and then having perceived that consumption a second later after stim ceased?

We did not observe rebound drinking in ChR2-expressing mice.

While both values are significant for Htr2a in Fig 3D-E, I am wondering if it would be beneficial to calculate Cohen's D here because it seems the effect size is bigger in the IC++ experiment. It is probably problematic to directly compare because of the 2 different opsins, but interesting nonetheless? Referring to IC++ as a "blue-shifted halorhodopsin" is not entirely accurate as one is an ion pump and one is a channel (though the activation wavelength difference is correct).

We thank the reviewer for the suggestion. Below is the calculation for the Cohen's D:

	Cohen's d
Htr2a CeA eNpHR3.0/Ctrl	1.081
Htr2a CeM IC++/Ctrl	1.949
Htr2a CeA eNpHR3.0/Htr2a CeM IC++	0.801

The effect size in the CeM^{Htr2a} versus the CeA^{Htr2a} neuron experiment, measured by Cohen's d, was 0.8, indicating a medium effect. We added this to the results part.

Additionally, we appreciate the correction regarding the IC++. We have now described it as blue-shifted chloride-conducting channelrhodopsin.

For the preference index in the RTPP experiments (Fig 4) please draw a dashed line at 50% so that the reader can have that context when looking at the data. Also it would be good to keep preference index either a % or a scale of 0-1 in the same figure.

We thank the reviewer for the suggestion. We modified the figure accordingly.

With regard to Fig 5 using the phrase “stimulating” (especially right after doing RTPP opto in Fig 4). Is somewhat confusing to the reader, perhaps there is a better way to phrase this?

We thank the reviewer for the suggestion. We made sure to always use ‘photoactivation’ in the context of optogenetic manipulation and to use the term ‘stimulus exposure’ in Fig. 5. For example:

“When comparing the activities of all active cells during habituation and stimulation (including active and inactive times), we observed a general increase in neuronal activity for both CeM^{Htr2a} and CeM^{Sst} neurons during **stimulus exposure**, except for CeM^{Htr2a} neurons exposed to water (Fig. 5c,d).”

With regard to the specializer vs. generalizer cells in Fig 6, it is slightly problematic that the authors did not use a more aversive dose of quinine. What tests were conducted to determine if 10 uM quinine was aversive? In previous studies following surgical manipulation it has been noted that there is a shift in the quinine DR curve, where concentrations 10 uM and lower did not result in aversion behavior (see Torruella-Suarez et al., 2020.) Was the tastant ordering counterbalanced for presentation?

We thank the reviewer for the comment. Perhaps, we did not state this clearly enough, but the quinine concentration, as mentioned in the Materials and Methods, was 10 mM, not 10 μ M. This is a rather high concentration, chosen specifically for its aversive properties and matching the concentration used in our previous work (Douglass et al., 2017).

As suggested by reviewer #3, we have included new photoactivation experiments in which water-deprived mice were exposed to a 10 mM quinine solution. While photoactivated control mice avoided the bitter solution, activation of CeM^{Htr2a} and CeM^{Sst} neurons stimulated quinine consumption. During subsequent days, we tested the same mice with a subtler quinine solution (100 μ M) under water-deprived conditions for 30 minutes and observed that photoactivated ChR2-expressing mice consumed more fluid compared to controls. Comparing the intake of the two quinine concentrations by the same mice, we found that controls consumed similar small amounts of both solutions, whereas ChR2-expressing mice drank significantly more of the lower-concentrated solution (new Figure 6i,j).

The tastant order was not counterbalanced.

Figure 8 is a tour de force and nicely brings together the main concepts of the paper. Still it would be interesting to see what would happen with the CeAL projection neurons to the PBN, especially given the differences in terminal expression within the PBN! I feel this experiment, maybe even if just completed for a limited set of endpoints would be valuable to the field. I would put the Sst images in the main figure

and not bury them in the supplement. Especially because the representative image for the PBN looks somewhat strange from the Sst CeM?

We thank the reviewer for these suggestions. We have moved the Sst images to the main Figure 8. In addition, we have quantified the projections by calculating the integrated fluorescence intensities normalized to background and injection sites. Images and quantifications of the major projections of CeM and CeL Htr2a and Sst neurons are now shown in the main Figure 8. All projection data are presented in two new Extended data figures 8 and 9. The results are interesting and discussed in the manuscript (lines 370-382, 532-541).

We agree that manipulations of the CeL projections to the PBN would have been a useful addition to the story. It is unlikely that activating CeL projection neurons alone would promote water or food intake, given that CeL cell body activation did not produce notable effects. It is possible that activating CeL projection neurons may be sufficient to drive conditioned reward behavior. However, with limited numbers of animals at the correct age, we had to weigh the benefits of those experiments against the requests of reviewer 3 for additional optogenetic experiments using quinine adulterated water. These experiments seemed important to disentangle fluid intake versus valence detection (see below and revised main figure 6).

Some of the interpretations maybe overstated and could be dialed back a bit. For example, the authors are trying to show that CeL neurons are not rewarding in RTPP but are in conditioned flavor preference – I think there might be a less strong effect (there appears to be trends in Fig 4C and single animals in the control and Chr2 conditions for Htr2a driving effects), but still a possible effect. They even say in the discussion that “the inferior performance of the Con/Foff virus may have resulted from incomplete coverage of Flp expression in CeL neurons.” Could this not be part of the explanation for why the trend looks the same, but isn’t as important as CeM. While it may be equally/more important for drinking than CeM, the authors may be over interpreting the differences here.

We agree and have discussed the system as being effective, but not perfect:

“Here, we have used a newly generated mouse line that expresses the optimized and tamoxifen-inducible FlpoER recombinase in the CeL subdivision of the CeA, to a much higher degree (17-fold) than in cells of the CeM. In combination with a specific Cre line and a Con/Fon Boolean reporter virus, we could target between 4- and 10-times more Cre+ cells in the CeL than CeM. Conversely, with a Con/Foff virus, we could target between 2.3- and 2.7-times more Cre+ cells in the CeM than CeL. These results indicate that the system is effective at producing an enrichment of CeL versus CeM subpopulations. The inferior performance of the Con/Foff virus may have resulted from incomplete coverage of Wfs1-Flp expression in CeL neurons. Cre-positive CeL cells that do not co-express Flp would express the Con/Foff reporter and thereby lower the CeM:CeL ratio. As for most biological systems, the expression in the CeL

versus CeM subpopulations is not black-and-white. The strength of the system relies on the combination of the INTRSECT viruses with photoactivation and -inhibition experiments.”

As for the specific example, we have dialed back our interpretation by stating: “The CeL subtypes of Sst+ and Htr2a-Cre-expressing neurons also displayed modest reinforcing activity in the conditioned flavor preference assay, but perhaps not in RTPP.” (Line 515-517)

Reviewer #3 (Remarks to the Author):

Summary

One of the hallmarks of CeA biology is the extensive heterogeneity of CeA neurones at multiple anatomical, gene expression, and physiological levels. The development of a FlpO driver to mark CeL neurones is a major and important advance to allow for intersectional targeting to characterize heterogeneity. The authors perform a number of important experiments to dissect essentially four mostly distinct and enriched populations of CeA neurones. The identification of new behaviors and encoding (particularly Fig. 5/6) is an important step forward for the field and this lab going back to the original findings in Dougless et al, NN. 2017. Overall, I like this paper and have made some suggestions below that I hope will improve things. In some cases these are dealbreakers. In others I spell out what are not dealbreakers, but in hopes that this will improve future experiments and ongoing projects.

Major revision points

Overall: The authors have done a heroic amount of work and made some important advances. I encourage them, in the future, to endeavour to run more powered studies where exploration of sex differences is possible and to plot their data in a way that allows the reader to appreciate potential sex differences. I recognize it would be a monumental amount of work to redo all of their figures in this way so it is not a requirement for my potential and ultimate acceptance of this manuscript. Nonetheless, I highly encourage the authors to do this for future studies...even if it means the experiments take longer. In the figure legends, please list the number of animals for each group. If it is the same for each panel, it can be done at the end of the legend. In some places it was hard to gauge the rigor of the experiments due to the omission of this information.

We thank the reviewer for the suggestions and recognition of our work. We agree that exploring sex differences is crucial and we will prioritize this in future studies. In response to the comment, we have plotted some behavioral data by separating males and females (Fig. 3 below). While the number of females in these experiments is insufficient to achieve statistical significance, we observed similar trends across both groups. Additionally, we have updated the figure legends to include the number of mice per

group.

Fig.3

- a.** Amount of water consumed by Htr2a-Cre;Wfs1-FlpoER water-deprived mice expressing ChR2 or EYFP in the CeM^{Htr2a} during 30 min photoactivation (Males: unpaired t test $p=0.0012$, $t=4.332$. Females: unpaired t test $p=0.0904$, $t=2.094$). (n = 7 Ctrl, n = 6 ChR2 males; n = 4 Ctrl, n = 3 ChR2 females).
- b.** Amount of water consumed by Sst-Cre;Wfs1-FlpoER water-deprived mice expressing ChR2 or EYFP in the CeM^{Sst} during 30 min photostimulation (Males: unpaired t test $p=0.0003$, $t=5.016$. Females: unpaired t test $p=0.1627$, $t=1.636$). (n = 6 Ctrl, n = 8 ChR2 males; n = 4 Ctrl, n = 3 ChR2 females).
- c.** Amount of water consumed by hydrated Htr2a-Cre;Wfs1-FlpoER mice expressing ChR2 or EYFP in the CeM^{Htr2a} during 30 min optogenetic activation (Males: Mann-Whitney U test $p=0.0006$, $U=0$. Females: Wilcoxon Signed Rank Test $p=0.1250$). (n = 7 Ctrl, n = 7 ChR2 males; n = 2 Ctrl, n = 4 ChR2 females).
- d.** Amount of water consumed by hydrated Sst-Cre;Wfs1-FlpoER mice expressing ChR2 or EYFP in the CeM^{Sst} during 30 min (Males: Mann-Whitney U test $p=0.0028$, $U=3.500$. Females: Wilcoxon Signed Rank Test $p=0.2500$). (n = 6 Ctrl, n = 9 ChR2 males; n = 4 Ctrl, n = 3 ChR2 females).
- e.** Amount of food consumed by Htr2a-Cre;Wfs1-FlpoER mice expressing ChR2 or EYFP in the CeM^{Htr2a} during 40 min photostimulation (Males: Mann-Whitney U test $p=0.0373$, $U=8$. Females: Mann-Whitney U test $p=0.5000$, $U=2.500$). (n = 7 Ctrl, n = 6 ChR2 males; n = 3 Ctrl, n = 3 ChR2 females).
- f.** Preference for the light-paired chamber when the entire population of CeM^{Htr2a} neurons is activated in the RTPP task, compared to controls (Males: unpaired t test $p=0.0022$, $t=3.970$. Females: unpaired t test $p=0.3455$, $t=1.068$). (n = 7 Ctrl, n = 6 ChR2 males; n = 3 Ctrl, n = 3 ChR2 females).
- g.** Preference for the light-paired chamber when the entire population of CeM^{Sst} neurons is activated in the RTPP task, compared to controls (Males: unpaired t test $p=0.1010$, $t=1.776$. Females: unpaired t test $p=0.5580$, $t=0.8325$). (n = 6 Ctrl, n = 8 ChR2 males; n = 2 Ctrl, n = 1 ChR2 females).
- h.** Preference for the light-paired chamber when the entire population of CeL^{Htr2a} neurons is activated in the RTPP task, compared to controls (Males: Mann-Whitney U test $p=0.0200$, $U=6$. Females: unpaired t test $p=0.8332$, $t=0.2219$). (n = 6 Ctrl, n = 8 ChR2 males; n = 3 Ctrl, n = 4 ChR2 females).
- i.** Preference for the light-paired chamber when the entire populations of CeL^{Sst} neurons is activated in the RTPP task, compared to controls (Males: unpaired t test $p=0.6381$, $t=0.4850$. Females: unpaired t test $p=0.4340$, $t=0.8299$). (n = 7 Ctrl, n = 5 ChR2 males; n = 5 Ctrl, n = 4 ChR2 females).

Figure 1: The intersectional targeting approach is effective at producing an enrichment of CeL vs CeM Htr2a and Sst neurones. It is not perfect. This needs to be discussed as a potential limitation in the Discussion section. Also it is possible that the Wfs1-FlpoER line is not penetrant in all Wfs1+ cells (Extended Data 1k-o) contributing to the CeM intersectional targeting being slightly less specific. This is mentioned at line 420 somewhat. This does not undermine the study. Any other reviewer that suggests this is missing the forest for the trees.

We thank the reviewer for their positive comments. We have discussed the potential limitation of the approach in the discussion.

“Here, we have used a newly generated mouse line that expresses the optimized and tamoxifen-inducible FlpoER recombinase in the CeL subdivision of the CeA, to a much higher degree (17-fold) than in cells of the CeM. In combination with a specific Cre line and a Con/Fon Boolean reporter virus, we could target between 4- and 10-times more Cre+ cells in the CeL than CeM. Conversely, with a Con/Foff virus, we could target between 2.3- and 2.7-times more Cre+ cells in the CeM than CeL. These results indicate that the system is effective at producing an enrichment of CeL versus CeM subpopulations. The inferior performance of the Con/Foff virus may have resulted from incomplete coverage of Wfs1-Flp expression in CeL neurons. Cre-positive CeL cells that do not co-express Flp would express the Con/Foff reporter and thereby lower the CeM:CeL ratio. As for most biological systems, the expression in the CeL versus CeM subpopulations is not black-and-white. The strength of the system relies on the combination of the INTRSECT viruses with photoactivation and -inhibition experiments.”

The revised manuscript now has the information on the exact chromosomal location of the two novel Flpo transgenic lines (methods).

Figure 2. An attempt to describe or depict the optical fiber placement of each mouse plotted should be shown here or in the extended data as well as an explanation of any animals removed from the data shown due to predetermined criteria. The data in this figure are very nice and impactful, but the rigor should be mentioned at some point. Also, no attempt was made to disentangle fluid vs caloricity vs palatability. This is not a dealbreaker, but promotion of fluid intake even in the face of subtle quinine adulteration would add that activation of these cells is stimulating a licking and fluid intake motor program or that its activation is capable of encoding valence for fluids. The conditioned taste experiment is consistent with this idea, but the lack of a negative flavor dissection in Fig. 2 is highlighted when it is included in Fig. 6. The data here suggests that Htr2a neurons may encode the valence of the fluid. Disentangling these possibilities with free consumption of quinine adulterated fluid with opto would be impactful.

We thank the reviewer for making these valuable suggestions. We prepared a schematic illustration to depict the placements of optical fibers in the brain of the mice and included histological images showing optical fiber positioning and EYFP expression in the CeA. The data is now shown in Extended data figure 2). We also prepared illustrations for GRIN lens placement in the CeA of animals expressing the GCaMP sensor and for optical fiber placement in the PBN (Extended data figures 7 and 10). As an example of an excluded animal, we added an image of a mouse in which viral expression could not be detected after the behavioral experiments (Extended data figure 2 g).

As suggested, we have generated new experimental data to disentangle fluid intake versus valence detection. We included both CeM^{Htr2a} and CeM^{Sst} subpopulations. Neurons were photoactivated during 30 min in which water-deprived mice were exposed to a 10 mM quinine solution. While photoactivated

control mice avoided the bitter solution, activation of CeM^{Htr2a} and CeM^{Sst} neurons stimulated quinine consumption. During subsequent days, we tested the same mice with a subtler quinine solution (100 μ M) under water-deprived conditions for 30 minutes and observed that photoactivated Chr2-expressing mice consumed more fluid compared to controls. Comparing the intake of the two quinine concentrations by the same mice, we found that controls consumed similar small amounts of both solutions, whereas Chr2-expressing mice drank significantly more of the lower-concentrated solution (new Figure 6i,j).

These findings suggest that photoactivation of CeM^{Htr2a} and CeM^{Sst} neurons stimulates a fluid intake motor program, but that the mice are still able to distinguish varying concentrations of unpleasant taste. Considering that the neurons detect positive and negative valence for fluids, as suggested by the calcium data, and that neuron activation is rewarding as indicated by RTPP and self-stimulation (Douglass et al., 2017), we conclude that photoactivation of the neurons has two effects that contribute to fluid consumption: First, the activation of a fluid intake motor program that drives drinking independent of internal state and despite strongly aversive taste, and second, the generation of a rewarding effect that counterbalances the bitter taste. The stronger the bitter taste, the more it cancels out the rewarding effects and the less is consumed by the mice. We have added this discussion to the main text (line 502-510).

Extended Data 4: Please show the distance traveled in the open field. Center time is difficult to interpret without it. If there is substantial variance, then the data can be shown as % of time spent in the center in which case Htr2a overall activation might trend towards anxiolysis but we can't determine this without distance traveled. Even if one of these cohorts did show anxiolysis recalculated as % time I don't think this undermines the study and is consistent with the RTPP. In other words, there is no reason not to show this.

We have updated the open field graphs to display the percentage of time spent in the center relative to the total time, providing a clearer interpretation of the data (Ext. data Fig. 5 a,b,c). Additionally, we have included new graphs showing the total distance traveled during the open field test (Ext. data Fig. 5 a,b,c). Upon analysis, we did not observe any statistically significant differences.

Figure 5 K,I and 6 F,G: Is there a way to highlight what neurons are specializers in the chord diagrams or in the side bar graphs? Also do the individual data points represent the % of neurons recorded from an individual animal? The reader is having to infer a lot from these plots and the text. It's not overly clear how these experiments were done, so more detail can be added to the methods as well. As with the opto data, the histology of GRIN lens implanted animals or rigor thereof should be shown or detailed somewhere.

We thank the reviewer for the comment. We have addressed the suggestion to highlight the specializers by adding a gray background behind the bar graphs belonging to the specializers. Regarding the individual data points in the bar graphs (Fig. 5 k,l, 6 f,g), these represent the percentage of neurons from each animal that maintained or changed their correlation across the two stimuli. This information was added to the figure legends. Additionally, we have included a representative histological image of the GRIN lens implanted in CeM^{Htr2a} and CeM^{Sst} animals (Extended data figure 7a).

Figure 7: The electrophysiological studies could be much improved in this figure with regards to how they are explained and executed. There is no assessment of spike fidelity or latency in PKCdelta neurones and the authors say “providing evidence for a monosynaptic connection”. There was no mention of drugs used to examine the specificity of optically induced IPSCs in the CeM (TTX, 4-AP) or blockade of the putative IPSC (GABAazine, picrotoxin). I agree that this all likely to be the case and perhaps unsurprising, but the rigor is lacking and data overstated. Please improve this.

We agree with the reviewer that TTX and 4-AP are normally used to evaluate monosynaptic connections. However, it is also true that when the neuron expressing channelrhodopsin is GABAergic, the use of these drugs is not necessary, as polysynaptic excitation will not occur. Therefore, we did not use these drugs. Nevertheless, considering the comments of the reviewer, we have changed the statement “providing evidence for a monosynaptic connection” for “suggesting evidence for a monosynaptic connection”. Regarding the blockade of the putative IPSC, for the recordings we have used a low chloride internal solution (K-gluconate) and patched the neurons with a voltage of 0 mV, allowing us to only observe inhibitory responses (IPSC) and not the excitatory (EPSC). We have previously used a similar strategy to evaluate the connections inside the CeA for CeA-Dlk1 neurons (Ding et al., Cell Reports, 2024) and therefore, we did not use drugs to differentiate between EPSC and IPSC. This information was added to the Materials and Methods section.

Figure 8: An average intensity calculation of the output terminals at each region of CeM vs CeL Htr2a and Sst neurons would provide quantification and support for areas in which the CeM “covered a larger area within the target region” (line 357). I’d also like to see each of the areas shown if possible. The PVT enrichment for CeM Htr2a neurones is particularly intriguing but I don’t see it in Figure 8 or Extended Data 7. As before, rigor of optical fiber placement should be shown or described (at least).

We thank the reviewer for the comment and suggestion to quantify the projections. This was done and the results are presented in revised Fig. 8 and Extended data Figures 8 and 9. We described the results as follows:

“To determine the relative strength of the projections, we calculated the integrated fluorescence intensities in the output region normalized to background and injection sites. Our analysis revealed that for Htr2a neurons, the CeM fraction projected to a larger number of outputs compared to the CeL fraction (19 versus 8), whereas for Sst neurons, CeM and CeL fractions projected to similar numbers of outputs (20 versus 16) (Extended data Figs. 8, 9). A number of brain regions received strong projections from both Htr2a and Sst CeL and CeM fractions, including the bed nucleus of the stria terminalis (BNST), the lateral vestibular nucleus (LAV), the lateral and medial parabrachial nucleus (LPBN, MPBN), and the midbrain reticular nucleus (MRN) (Fig. 8a-d). The interstitial nucleus of the posterior limb of the anterior commissure (IPAC) and the substantia innominata (SI) received stronger projections from the CeM compared to the CeL fractions. Few brain regions received enriched projections from CeM^{Sst} neurons, including the lateral and medial geniculate complex (LG, MG) and the ventral posterolateral/medial nucleus of the thalamus (VPL/M) (Fig. 8 a-d; Extended data Fig. 8, 9). Several of these regions are known to be involved in rewarding and consummatory behaviors.”

The discussion reads as follows:

A number of brain regions received strong projections from both Htr2a and Sst CeL and CeM fractions. Among them, the BNST plays a central role in reward-related behaviors⁵⁹, while also influencing feeding through its projections to the lateral hypothalamus (LH)^{60,61}. The parabrachial nucleus is a sensory relay receiving an array of interoceptive and exteroceptive inputs relevant to taste and ingestive behavior, pain, and multiple aspects of autonomic control^{39,45,62-64}. Brain regions that receive stronger projections from CeM than CeL neurons include the IPAC, located within the extended amygdala, and known to regulate energy homeostasis⁶⁵ and the SI, a basal forebrain structure involved in reinforcement learning⁶⁶. Few brain regions received enriched projections from CeM^{Sst} neurons. Among them, the lateral and medial geniculate complex (LG, MG), thalamic nodes involved in defensive behaviors, and the ventral posterolateral/medial nucleus of the thalamus (VPL/M), a relay for somatosensory signals⁶⁷⁻⁷¹.

Minor revision points

Title: Why “uptake” instead of “intake”? The former is weird. Animals don’t “take up” food or water, they “take in” food or water. This is used throughout the manuscript and it is distracting.

We agree and introduced the suggested changes.

Introduction:

Line 47: CeA neurones can also be segregated and classified based on their physiological properties. See work from Yamar Carrasquillo.

Line 66-67: "fully worked out". Avoid colloquial phrases.

We added the work from Yamar Carrasquillo (Adke et al., 2021) and changed the text accordingly (line 47).

Line 82: "Unexpectedly". I understand using narrative adverbs to tell the story, but this is not unexpected in my opinion. Ip et al. Cell Metabolism 2019 (<https://pubmed.ncbi.nlm.nih.gov/31031093/>) was not cited. This is an important paper in adding to the idea that CeA neurones can actually promote feeding. Moreover, the localization of CeA NPY neurones to the CeM subdivision is very important to the overall conclusions of this study. It should be cited and discussed. This does not undercut the substantive impact and rigor of the current study to dissect CeL vs CeM subpopulations.

We thank the reviewer for recognizing this important omission. We have added the reference and discussed the findings in the revised manuscript (line 457-458).

Results:

Line 170: Food "uptake".

Line 232: neuronal "activities". Neuronal "activity" is better. Multiple instances of this throughout the results section.

Line 238: "relation"

We changed the text as suggested.

Line 366: This whole paragraph should be moved to the Discussion section.

Agreed and done as suggested.

Discussion: Fine

Methods: Fine, but see points above.

Overall decision: Please submit a revised manuscript for a second review addressing some of these

comments.